



# Modelling the climate and surface mass balance of polar ice sheets using RACMO2, Part 1: Greenland (1958-2016)

Brice Noël[1], Willem Jan van de Berg[1], J. Melchior van Wessem[1], Erik van Meijgaard[2], Dirk van As[3], Jan T. M. Lenaerts[4], Stef Lhermitte[5], Peter Kuipers Munneke[1], C. J. P. Paul Smeets[1], Lambertus H. van Ulft[2], Roderik S. W. van de Wal[1], and Michiel R. van den Broeke[1]

[1]Institute for Marine and Atmospheric research Utrecht, University of Utrecht, Utrecht, Netherlands.
[2]Royal Netherlands Meteorological Institute, De Bilt, Netherlands.
[3]Geological Survey of Denmark and Greenland (GEUS), Copenhagen, Denmark.
[4]Department of Atmospheric and Oceanic Sciences, University of Colorado, Boulder, USA.
[5]Department of Geoscience & Remote Sensing, Delft University of Technology, Delft, Netherlands.

*Correspondence to:* Brice Noël (B.P.Y.Noel@uu.nl)

**Abstract.**

We evaluate modelled Greenland ice sheet (GrIS) near-surface climate, surface energy balance (SEB) and surface mass balance (SMB) from the updated regional climate model RACMO2 (1958-2016). The new model version, referred to as RACMO2.3p2, incorporates updated glacier outlines, topography and ice albedo fields. Parameters in the cloud scheme governing the conversion of cloud condensate into precipitation have been tuned to correct inland snowfall underestimation; snow properties are modified to reduce drifting snow and melt production in the ice sheet percolation zone. The ice albedo prescribed in the updated model is lower at the ice sheet margins, increasing ice melt locally. RACMO2.3p2 shows good agreement compared to in situ meteorological data and point SEB/SMB measurements, and better resolves SMB patterns than the previous model version, notably in the northeast, southeast, and along the K-transect in southwestern Greenland. This new model version provides updated, high-resolution gridded fields of the GrIS present-day climate and SMB, and will be used for future climate scenario projections in a forthcoming study.



## 1 Introduction

Predicting future mass changes of the Greenland ice sheet (GrIS) using regional climate models
(RCMs) remains challenging (Rae et al., 2012). The reliability of projections depend on the ability
of RCMs to reproduce the contemporary GrIS climate and surface mass balance (SMB), i.e. snowfall
accumulation minus ablation from meltwater runoff, sublimation and drifting snow erosion (Van
Angelen et al., 2013a; Fettweis et al., 2013). In addition, model simulations are affected by the

quality of the re-analysis used as lateral forcing (Fettweis et al., 2013, 2017; Bromwich et al., 2015)
and by the accuracy of the ice sheet mask and topography prescribed in models (Vernon et al., 2013).

Besides direct RCM simulations, the contemporary SMB of the GrIS has been reconstructed using
various other methods, e.g. Positive Degree Day (PDD) models forced by statistically downscaled re-
analyses (Hanna et al., 2011; Wilton et al., 2016), mass balance models forced by the climatological

output of an RCM (HIRHAM4) (Mernild et al., 2010, 2011), and data assimilation from an RCM
combined with temperature and ice core accumulation measurements (Box, 2013). In addition,
Vizcaíno et al. (2013) used the Community Earth System Model (CESM) at $1°$ resolution ($\sim$100
km) to estimate recent and future mass losses of the GrIS.

Polar RCMs have the advantage to explicitly resolve the relevant atmospheric and surface physical

processes at high spatial (5 to 20 km) and temporal (sub-daily) resolution. Nonetheless, good RCM
performance often results from compensating errors between poorly parameterized processes, e.g.
cloud physics (Van Tricht et al., 2016) and turbulent fluxes (Noël et al., 2015; Fausto et al., 2016).
Therefore, considerable efforts have been dedicated to evaluate and improve polar RCM output in
Greenland (Ettema et al., 2010b; Van Angelen et al., 2013b; Lucas-Picher et al., 2012; Fettweis

et al., 2017; Noël et al., 2015; Langen et al., 2017), using in situ SMB observations (Bales et al.,
2001, 2009; Van de Wal et al., 2012; Machguth et al., 2016), airborne radar measurements of snow
accumulation (Koenig et al., 2016; Overly et al., 2016; Lewis et al., 2017) and meteorological records
(Ahlstrøm et al., 2008; Kuipers Munneke et al., 2017; Smeets et al., 2017), including radiative fluxes
that are required to close the ice sheet surface energy balance (SEB), and hence quantify surface

melt energy.

For more than two decades, the polar version of the Regional Atmospheric Climate Model (RACMO2)
has been developed to simulate the climate and SMB of the Antarctic and Greenland ice sheets. In
previous versions, snowfall accumulation was systematically underestimated in the GrIS interior,
while melt was generally overestimated in the percolation zone (Noël et al., 2015). At the ice sheet

margins, meltwater runoff is underestimated over narrow ablation zones and small outlet glaciers
that are not accurately resolved in the model's ice mask at 11 km. Locally, this underestimation can
exceed several m w.e. $\mathrm{yr}^{-1}$, e.g. at automatic weather station (AWS) QAS_L installed at the south-
ern tip of Greenland (Fausto et al., 2016). These biases can be significantly reduced by statistically
downscaling SMB components to 1 km resolution (Noël et al., 2016). Computational limitations

currently hamper direct near-kilometre simulations of the contemporary GrIS climate, making it





essential to further develop RACMO2 model physics at coarser spatial resolution.

Here, we present updated simulations of the contemporary GrIS climate and SMB at 11 km resolution (1958-2016). The updated model incorporates multiple adjustments, notably in the cloud scheme and snow module. Model evaluation is performed using in situ meteorological data and
point SEB/SMB measurements collected all over Greenland. We then compare the SMB of the updated model version (RACMO2.3p2) with its predecessor (RACMO2.3p1) for the overlapping period between the two simulations (1958-2015). Section 2 discusses the new model settings and initialisation, together with observational data used for model evaluation. Modelled climate and SEB components are evaluated using in situ measurements in Section 3. Changes in SMB patterns
between the new and old model versions are discussed in Section 4, as well as case studies in northeast, southwest and southeast Greenland. Section 5 introduces and evaluates the updated downscaled daily, 1 km SMB product. Section 6 discusses the remaining model uncertainties, followed by conclusions in Section 7. This manuscript is part of a tandem model evaluation over the Greenland (present study) and Antarctic ice sheets (Van Wessem et al., 2017).

## 2  Model and observational data

### 2.1  The Regional Atmospheric Climate Model RACMO2

The polar ('p') version of the Regional Atmospheric Climate Model (RACMO2) (Van Meijgaard et al., 2008) is specifically adapted to simulate the climate of polar ice sheets. The model incorporates the dynamical core of the High Resolution Limited Area Model (HIRLAM) (Undèn et al., 2002) and
the physics package cycle CY33r1 of the European Centre for Medium-range Weather Forecasts Integrated Forecast System (ECMWF-IFS, 2008). It also includes a multi-layer snow module that simulates melt, liquid water percolation and retention, refreezing and runoff (Ettema et al., 2010a), and accounts for dry snow densification following Ligtenberg et al. (2011). RACMO2 implements an albedo scheme that calculates snow albedo based on prognostic snow grain size, cloud optical
thickness, solar zenith angle and impurity concentration in snow (Kuipers Munneke et al., 2011). In RACMO2, impurity concentration, i.e soot, is prescribed as constant in time and space. The model also simulates drifting snow erosion and sublimation following Lenaerts et al. (2012a). Previously, RACMO2 has been used to reconstruct the contemporary SMB of the Greenland ice sheet (Van Angelen et al., 2013a,b; Noël et al., 2015, 2016) and peripheral ice caps (Noël et al., 2017a), the
Canadian Arctic Archipelago (Lenaerts et al., 2013; Noël et al., 2017b), Patagonia (Lenaerts et al., 2014) and Antarctica (Van Wessem et al., 2014a,b).



### 2.2 Surface energy budget and surface mass balance

In RACMO2, the excess energy obtained after closing the surface energy budget (SEB) is used to melt snow and ice (M) at the GrIS surface:

$$
\begin{aligned}
M &= SW_d + SW_u + LW_d + LW_u + SHF + LHF + G_s \\
&= SW_n + LW_n + SHF + LHF + G_s
\end{aligned}
\tag{1}
$$

where $SW_d$ and $SW_u$ are the shortwave down/upward radiation fluxes, $LW_d$ and $LW_u$ are the longwave down/upward radiation fluxes, SHF and LHF are the sensible and latent turbulent heat fluxes, and $G_s$ is the subsurface heat flux. $SW_n$ and $LW_n$ are the net short/longwave radiation at the surface. All fluxes are expressed in W m$^{-2}$ and are defined positive when directed towards the surface.

In the percolation zone of the GrIS, liquid water mass from melt (ME) and rainfall (RA) can percolate through the firn column, and is either retained by capillary forces as irreducible water (RT) or refreezes (RF). Combined with dry snow densification, this progressively depletes firn pore space until the entire column turns into ice (900 kg m$^{-3}$). The fraction not retained is assumed to immediately run off (RU) to the ocean:

$$
RU = ME + RA - RT - RF
\tag{2}
$$

The climatic mass balance (Cogley et al., 2011), hereafter referred to as SMB, is estimated as:

$$
SMB = P_{tot} - RU - SU_{tot} - ER_{ds}
\tag{3}
$$

where $P_{tot}$ is the total amount of precipitation, i.e. solid and liquid, RU is meltwater runoff, $SU_{tot}$ is the total sublimation from drifting snow and surface processes, and $ER_{ds}$ is the erosion by the process of drifting snow. All SMB components are expressed in mm w.e. (water equivalent) for point 'specific' SMB values, or in Gt yr$^{-1}$ when integrated over the GrIS.

### 2.3 Model updates

In the cloud scheme, parameters controlling precipitation formation have been modified to reduce the negative snowfall bias in the GrIS interior ($\sim$40 mm w.e. yr$^{-1}$) (Noël et al., 2015). To correct for this, the critical cloud content ($l_{crit}$) governing the onset of effective precipitation formation for liquid-mixed and ice clouds has been increased by a factor 2 (Eqs. 5.35 and 6.39 in ECMWF-IFS (2008)) and 5 (Eq. 6.42 in ECMWF-IFS (2008)), respectively. As a result, moisture transport is prolonged to higher elevations and precipitation is generated further inland.





Furthermore, the previous model version overestimated snow melt in the percolation zone of the
GrIS (Noël et al., 2015). With the aim of minimizing this bias, the following parameters have been
tuned in the snow module:

a) The model soot concentration, accounting for dust and black carbon impurities deposited on
snow, has been reduced from 0.1 ppmv to 0.05 ppmv, more representative of observed values (Do-
herty et al., 2010). A lower soot concentration yields a higher surface albedo, hence decreasing melt
(Van Angelen et al., 2012).

b) The size of refrozen snow grains has been reduced from 2 to 1 mm (Kuipers Munneke et al.,
2011). Consequently, the surface albedo of refrozen snow increases, as smaller particles enhance
scattering of solar radiation back to the atmosphere (Kaasalainen et al., 2006).

c) In previous model versions, the albedo of superimposed ice, i.e. the frozen crust forming at
the firn surface, was set equal to the albedo of bare ice ($\sim$0.55), underestimating surface albedo
and hence overestimating melt. The snow albedo scheme now explicitly calculates the albedo of
superimposed ice layers ($\sim$0.75), following Kuipers Munneke et al. (2011).

d) The saltation coefficient of drifting snow has been approximately halved from 0.385 to 0.190
(Lenaerts et al., 2012a). Saltation occurs when near surface wind speed is sufficiently high to lift
snow grains from the surface. In RACMO2, this coefficient determines the depth of the saltation
layer, i.e. typically extending 0 to 10 cm above the surface, that directly controls the mass of drifting
snow transported in the suspension layer aloft (above 10 cm). This revision does not affect the timing
and frequency of drifting snow events, which are well modelled (Lenaerts et al., 2012a,b), but only
reduces the horizontal drifting snow transport and sublimation, preventing a too early exposure of
bare ice during the melt season, especially in the dry and windy northeastern GrIS (Section 4.2).

**2.4 Initialisation and set up**

To enable a direct comparison with previous runs, RACMO2.3p2 is run at 11 km horizontal resolu-
tion for the period 1958-2016, and is forced at its lateral boundaries by ERA-40 (1958-1978) (Up-
pala et al., 2005) and ERA-Interim (1979-2016) (Dee et al., 2011) re-analyses on a 6-hourly basis
(Fig. 1). The forcing consists of temperature, specific humidity, pressure, wind speed and direction
being prescribed at each of the 40 vertical atmosphere hybrid model levels. Upper atmosphere re-
laxation (nudging) is also implemented in this new model version (Van de Berg and Medley, 2016).
As the model does not incorporate a dedicated ocean module, sea surface temperature and sea ice
cover are prescribed from the re-analyses (Stark et al., 2007). The model has about 40 active snow
layers that are initialised in September 1957 using the best temperature and density profile estimates
derived from the offline IMAU Firn Densification Model (IMAU-FDM) (Ligtenberg et al., 2011).
The data spanning the winter season up to December 1957 serve as an additional spin up for the
snowpack and are therefore discarded in the present study.

Relative to previous versions, the integration domain extends further to the west, north and east



(Fig. 1). This brings the northernmost sectors of the Canadian Arctic Archipelago and Svalbard well inside the domain interior, and further away from the lateral boundary relaxation zone (24 grid cells, black dots in Fig. 1). In addition, RACMO2.3p2 utilises the 90-m Greenland Ice Mapping Project (GIMP) Digital Elevation Model (DEM) (Howat et al., 2014) to better represent the glacier outlines and the surface topography of the GrIS. Compared to the previous model version, which

used the 5 km DEM presented in Bamber et al. (2001), the GrIS area is reduced by 10,000 km$^2$ (Fig. 2a). This mainly results from an improved partitioning between the ice sheet and peripheral ice caps, for which the ice-covered area has, in equal amounts, decreased and increased, respectively. The updated topography shows significant differences compared to the previous version, especially over marginal outlet glaciers where surface elevation has considerably decreased (Fig. 2b). Bare ice

albedo is prescribed from the 500 m MODerate-resolution Imaging Spectroradiometer (MODIS) 16-day Albedo product (MCD43A3), as the 5% lowest surface albedo records for the period 2000-2015 (vs. 2001-2010 in older versions; Fig. 2c). In RACMO2, ice albedo is minimized at 0.30 for dark ice in the low-lying ablation zone, and maximized at 0.55 for bright ice under perennial snow cover in the accumulation zone. In previous RACMO2 versions, bare ice albedo of glaciated grid cells

without valid MODIS estimate were set to 0.47 (Noël et al., 2015).

## 2.5  Observational data

To evaluate the modelled contemporary climate and SMB of the GrIS, we use daily average meteorological records of near-surface temperature, wind speed, relative humidity, air pressure and down/upward short/longwave radiative fluxes, retrieved from 23 AWS for the period 2004-2016

(green dots in Fig. 1). Erroneous radiation measurements, caused e.g. by sensor riming, were discarded by removing daily records showing $SW_{d\,bias} > 6\,\sigma_{bias}$, where $SW_{d\,bias}$ is the difference between daily modelled and observed $SW_d$ and $\sigma_{bias}$ is the standard deviation of the daily $SW_d$ bias for all measurements. In addition, measurements affected by sensor heating in summer, i.e. showing $LW_u > 318$ W m$^{-2}$, were eliminated as these values represent $T_s > 0°C$ for $\epsilon \approx 0.99$, where $T_s$ is

the surface temperature and $\epsilon$ the selected emissivity of snow or ice. We only used daily records that were simultaneously available for each of the four radiative components. Eighteen of these AWS sites are operated as part of the Programme for Monitoring of the Greenland Ice Sheet (PROMICE, www.promice.dk) covering the period 2007-2016 (Van As et al., 2011). Four other AWS sites, namely S5, S6, S9 and S10 (2004-2016), are located along the K-transect in southwest Greenland

(67°N, 47-50°W) (Smeets et al., 2017). Another AWS (2014-2016) is situated in southeast Greenland (66°N; 33°W) at a firn aquifer site (Forster et al., 2014; Koenig et al., 2014). The latter five sites are operated by the Institute for Marine and Atmospheric research at Utrecht University (IMAU).

We also use in situ SMB measurements collected at 213 stake sites in the GrIS ablation zone (yellow dots in Fig. 1; Machguth et al. (2016)) and at 182 sites in the accumulation zone (white dots

in Fig. 1) including snow pits, firn cores (Bales et al., 2001, 2009), and airborne radar measurements




(Overly et al., 2016). We exclusively selected measurements that temporally overlap with the model simulation (1958-2016). To match the observational period, daily modelled SMB is cumulated for the exact number of measuring days at each site.

For model evaluation, we select the grid cell nearest to the observation site in the accumulation zone. In the ablation zone, an additional altitude correction is applied by selecting the model grid cell with the smallest elevation bias among the nearest grid cell and its eight adjacent neighbours. One ablation site and seven PROMICE AWS sites presented an elevation bias in excess of $> 100$ m compared to the model topography and were discarded from the comparison.

### 3   Results: near-surface climate and SEB

We evaluate the modelled present-day near-surface climate of the GrIS in RACMO2.3p2 using data of 23 AWS sites (see Section 2.5). Then, we discuss in more detail the model performance at 4 AWS along the K-transect and compare RACMO2.3p2 output to those of RACMO2.3p1.

#### 3.1   Near-surface meteorology

Figure 3 compares daily mean values of 2-m temperature, 2-m specific humidity, 10-m wind speed,
air pressure collected at 23 AWS sites with RACMO2.3p2 output. The modelled 2-m temperature is in good agreement with observations ($R^2 = 0.95$) and with a RMSE of $\sim2.4°$C and a small cold bias of $\sim0.1°$C (Fig. 3a). As specific humidity is not directly measured at AWS sites, it is calculated from measured temperature, pressure and relative humidity following Curry and Webster (1999). The obtained 2-m specific humidity is accurately reproduced in the model ($R^2 = 0.95$) with a RMSE
$\sim0.35$ g kg$^{-1}$ and a negative bias of 0.13 g kg$^{-1}$ (Fig. 3b). The same holds for daily records of 10-m wind speed ($R^2 = 0.68$; Fig. 3c), with a small negative bias and RMSE of $\sim2$ m s$^{-1}$. Surface pressure is also well represented ($R^2 = 0.99$) with a small negative bias of 0.8 hPa and RMSE $< 6$ hPa (Fig. 3d). A systematic pressure bias at some stations results from the (uncorrected) elevation difference with respect to the model, which can be as large as 100 m.

#### 3.2   Radiative fluxes

Figure 4 shows scatter plots of modelled and measured daily mean radiative fluxes, i.e. short/longwave down/upward radiation. Radiative fluxes are also well reproduced by the model with $R^2$ ranging from 0.83 for LW$_d$ to 0.95 for SW$_d$ (Fig. 4), showing relatively small biases of -7.1 W m$^{-2}$ and 3.8 W m$^{-2}$, and RMSE of 21.2 W m$^{-2}$ and 27.1 W m$^{-2}$, respectively. The negative bias in LW$_d$, hence
leading to LW$_u$ underestimation of 4.4 W m$^{-2}$ with a small RMSE of 12.1 W m$^{-2}$, in combination with positive bias in SW$_d$ suggests an underestimation of cloud cover in the ice sheet marginal regions, where most stations are located. The larger bias and RMSE in SW$_u$ of 6.8 W m$^{-2}$ and 32.1 W m$^{-2}$, respectively, can be ascribed to overestimated surface albedo, especially during summer



snowfall episodes, when a bright fresh snow cover is deposited over bare ice. Note that these AWS

radiation measurements are also prone to potentially large uncertainties due to preferred location on
ice hills, sensor tilt, riming and snow/rain deposition on the instruments, leading to spurious albedo
and $SW_u$ data, e.g. the upper left dots in Fig. 4b.

### 3.3  Seasonal SEB cycle along the K-transect

The K-transect comprises four AWS sites located in different regions of the GrIS: S5 and S6 are

installed in the lower and upper ablation zone, respectively, S9 is situated close to the equilibrium
line and S10 in the accumulation zone. Figure 5 shows monthly mean modelled (continuous lines,
RACMO2.3p2) and observed (dashed lines) SEB components, i.e. net short/longwave radiation
($SW_n$/$LW_n$), latent and sensible heat fluxes (LHF/SHF), surface albedo and melt measured at these
four AWS sites for the period 2004-2015. Tables 1-4 list statistics calculated at each individual AWS

and for the two model versions.

#### 3.3.1  Low ablation zone

At station S5 (490 m a.s.l.), surface melt is well reproduced in RACMO2.3p2, with a small negative
bias of 0.4 W m$^{-2}$ (Table 1; Fig. 5b). However, this good agreement results from significant error
compensation between overestimated $SW_n$ (16.2 W m$^{-2}$) and underestimated SHF in summer (15.3

W m$^{-2}$; Fig. 5a). The bias in $SW_n$ is mostly driven by overestimated $SW_d$ (20.7 W m$^{-2}$; Table 1)
and to a lesser extent by $SW_u$ (4.5 W m$^{-2}$), resulting from too low cloud cover and ice albedo
(Fig. 5b), respectively. AWS are often installed on snow covered promontories, i.e. hummocks,
that maintain higher albedo in summer (∼0.55) than their surroundings where impurities collect.
Mixed reflectance from bright ice cover (∼0.55) and neighbouring darker tundra, exposed nunataks

or meltwater ponds (< 0.30), located within the same MODIS grid cell, likely explains this under-
estimation. Another explanation stems from the deterioration of MODIS sensors in time, resulting
in underestimated surface albedo records (Polashenski et al., 2015; Casey et al., 2017).

LW$_n$ is well reproduced in the model due to similar negative biases in LW$_d$ and LW$_u$ (∼12 W
m$^{-2}$), indicating again too low cloud cover. The large negative bias in SHF is attributed to an

inaccurate representation of surface roughness in the lowest sectors of the ablation zone. Smeets and
Van den Broeke (2008) show that observed surface roughness for momentum has a high temporal
variability at site S5, with a minimum of 0.1 mm in winter, when a smooth snow layer covers the
rugged ice sheet topography, and a peak in summer (up to 50 mm), when melting snow exposes
hummocky ice at the surface. In RACMO2, surface aerodynamic roughness is prescribed at 1 mm

for snow-covered grid cells and at 5 mm for bare ice, hence significantly underestimating values
over ice in summer and thus causing too low SHF (Ettema et al., 2010a). This bias in SHF at S5 is
also partly ascribable to too cold conditions (2°C). Although not negligible, LHF contributes little
to the energy budget and shows a positive bias of ∼3 W m$^{-2}$, notably in winter.



### 3.3.2 Upper ablation zone

Station S6 is located at 1010 m a.s.l. in the GrIS upper ablation zone. There, summer melt is overestimated by $\sim$8 W m$^{-2}$ owing to both too high SW$_n$ and SHF (2.2 W m$^{-2}$ and 7 W m$^{-2}$, respectively; Fig. 5c). As for S5, the bias in SW$_n$ results from overestimated SW$_d$ (6 W m$^{-2}$) and underestimated SW$_u$ (4 W m$^{-2}$). At the AWS location, surface albedo progressively declines from 0.60 to $\sim$0.40 when bare ice is exposed in late summer, whereas RACMO2.3p2 simulates bare ice

at the surface throughout summer, with an albedo of 0.40. As a result, modelled surface albedo is systematically underestimated in summer, especially in July (Fig. 5d). Likewise, a small negative bias in LW$_n$ ($\sim$2 W m$^{-2}$) is obtained as LW$_d$ and LW$_u$ are both slightly underestimated (Table 2). Here, 2-m temperature is on average 0.7°C too high, causing a too large SHF (7 W m$^{-2}$).

### 3.3.3 Equilibrium line

Close to the equilibrium line, RACMO2.3p2 slightly underestimates summer melt (2.4 W m$^{-2}$; Fig. 5f and Table 3). At station S9 (1520 m a.s.l.), a perennial snow cover maintains a minimum albedo of 0.65 in summer, i.e. when melt wets the snow. A small positive bias in modelled snow albedo (0.03) combined with a slightly underestimated SW$_d$ (1.5 W m$^{-2}$) lead to an overestimated SW$_u$ (3.5 W m$^{-2}$), hence underestimating SW$_n$ ($\sim$5 W m$^{-2}$). Although LW$_d$ and LW$_u$ are over-

estimated, especially in winter (3.1 W m$^{-2}$ and 0.5 W m$^{-2}$, respectively), LW$_n$ agrees well with measurements. The 2-m surface temperature shows a 0.5°C positive bias, in turn causing slightly too large SHF ($\sim$5 W m$^{-2}$; Fig. 5e and Table 3).

### 3.3.4 Accumulation zone

All SEB components are well reproduced at site S10 (1850 m a.s.l.). Compensation of minor errors

between underestimated SW$_d$ and overestimated SW$_u$ ($\sim$1 W m$^{-2}$) provides a good agreement with observed SW$_n$ (Fig. 5g). Modelled surface albedo also compares well with measurements, with only a small positive bias (0.03; Fig. 5h). LW$_n$ is underestimated by $\sim$5 W m$^{-2}$; this is mainly driven by a too low LW$_d$ and a too large LW$_u$ (Table 4). The turbulent fluxes are well captured although a significant bias in SHF persists ($\sim$7 W m$^{-2}$), especially in winter when LW$_d$ is underestimated. As

biases in SHF and LW$_d$ are almost equal, modelled melt matches well with observations despite a small negative bias ($\sim$2 W m$^{-2}$).

### 3.4 Model comparison along the K-transect

Tables 1-4 compare statistics of SEB components between RACMO2.3p2 and 2.3p1. Although differences are relatively small, the new model formulation shows general improvements. The in-

creased cloud cover over the GrIS reduced the bias in SW$_d$ and LW$_d$. Improvements in the representation of turbulent fluxes is partly attributed to the new topography prescribed in RACMO2.3p2



and the better resolved $SW_d$/$LW_d$, although significant biases remain at all stations.

At site S5 located in the low ablation zone (Table 1), smaller $SW_d$ and lower ice albedo significantly reduce the $SW_u$ bias in RACMO2.3p2, and enhanced $LW_d$ decreases the negative bias in $LW_u$. As a result, melt increases substantially, reducing the negative bias compared to version 2.3p1. Note that $SW_d$ remains overestimated in RACMO2.3p2. This is compensated by underestimated SHF, i.e. partly caused by underestimated $LW_d$, providing realistic surface melt. In the upper ablation zone, similar improvements are obtained at site S6 (Table 2). Here, all SEB components show smaller biases except for $SW_u$, as underestimated surface albedo increases the negative $SW_u$ bias.

Above the equilibrium line, enhanced cloud cover also reduces the SW and LW biases at sites S9 and S10 (Tabs. 3 and 4). However, surface albedo overestimation in RACMO2.3p2 causes a small increase in melt underestimation.

## 4 Results: regional SMB

In Section 3, we discussed the overall good ability of RACMO2.3p2 to reproduce the contemporary climate of the GrIS, which is essential to estimate realistic SMB patterns. Here, we first compare SMB of the new and old model over the GrIS. For further evaluation, we zoom in on three regions where large SMB differences exist between the two versions.

### 4.1 Changes in SMB patterns

Figure 6a shows SMB from RACMO2.3p2 for the overlapping model period 1958-2015. Differences with the previous version 2.3p1 are shown in Fig. 6b and the changes in individual SMB components are depicted in Fig. 7. Owing to the modifications in the cloud scheme, clouds are sustained to higher elevations, enhancing precipitation further inland, while it decreases in low-lying regions. Changes are especially large in southeast Greenland where the decrease locally exceeds 300 mm w.e. $yr^{-1}$. Precipitation in the interior increases by up to 50 mm w.e. $yr^{-1}$ (Fig. 7a). This pattern of change is clearly recognisable in the SMB difference (Fig. 6b). In addition, the shallower saltation layer in the revised drifting snow scheme is responsible for reduced sublimation ($\sim$50 mm w.e. $yr^{-1}$; Fig. 7b) that reinforces the overall increase in SMB (Fig. 6b). Although drifting snow erosion changes locally, patterns are heterogeneous and the changes remain small when integrated over the GrIS (Fig. 7c). This process has only a limited contribution to SMB ($\sim$1 Gt $yr^{-1}$) resulting from drifting snow being transported away from the ice sheet towards the ice-free tundra and ocean.

In the percolation zone, the decrease in runoff (Fig. 7d) is governed by reduced surface melt (Fig. 7e), mostly resulting from the smaller grain size of refrozen snow and the lower soot concentration in snow that have increased surface albedo (not shown), further increasing SMB (Fig. 6b). In west and northeast Greenland, this decrease in runoff even exceeds that of melt by 50 to 100 mm





w.e. yr$^{-1}$, a result of enhanced precipitation that increased the snow refreezing capacity (Fig. 7f). At higher elevations, the decrease in refreezing is exclusively driven by melt reduction (Figs. 7e and f), while at the very GrIS margins, the lower ice albedo used in RACMO2.3p2 (Fig. 2c) locally increases runoff (Fig. 7d), in turn decreasing SMB (Fig. 6b).

### 4.2 Northeast Greenland

For northeast Greenland's two main glaciers, Zachariae Isstrøm and Nioghalvfjerdsbrae (79N glacier; yellow line in Fig. 6a), solid ice discharge estimates are available for the period 1975-2015 (Mouginot et al., 2015). In these two catchments, model updates significantly improve the representation of SMB, that was substantially underestimated in the previous version. Figure 8a compares ice dis-

charge (black dots) with modelled SMB (RACMO2.3p2 as blue dots and 2.3p1 in red) integrated over the two glacier basins for 1958-2015. In a balanced system, i.e. before discharge accelerated in 2001, SMB equals ice discharge. Averaged over 1975-2001, modelled SMB in RACMO2.3p2 (20.5 Gt yr$^{-1}$) is similar to the estimated glacial discharge of 21.2 Gt yr$^{-1}$, significantly improving upon version 2.3p1 (15.8 Gt yr$^{-1}$). The negative bias in RACMO2.3p2 (0.7 Gt yr$^{-1}$; dashed blue line)

is reduced by almost a factor of eight relative to the previous version (5.4 Gt yr$^{-1}$) and SMB now equals discharge within the uncertainty. Averaged over 2001-2015, basin mass loss accelerated due to enhanced surface runoff, decreasing SMB by 4.2 Gt yr$^{-1}$, and increased ice discharge (2.8 Gt yr$^{-1}$).

Figures 8b and c show mean SMB for 1958-2015 as modelled by RACMO2.3p2 and 2.3p1, re-

spectively. In the percolation zone, the difference between the two model versions primarily results from the smaller refrozen snow grain size that reduces melt and runoff through increased surface albedo in RACMO2.3p2. To a smaller extent, reduced soot concentration delays the onset of melt in summer. In the ablation zone, snow cover persists longer before bare ice is exposed in late summer, in turn reducing runoff (Fig. 7d). Superimposed on this, precipitation has increased over the

whole glacier basin (Fig. 7a), allowing for enhanced refreezing in snow (Fig. 7f) hence increasing SMB by 4.7 Gt yr$^{-1}$ in RACMO2.3p2 (Fig. 6b). Note the large inter-annual variability in modelled SMB showing a maximum and minimum value of approximately 30 Gt yr$^{-1}$ and 8.5 Gt yr$^{-1}$ in RACMO2.3p2 vs. 25 Gt yr$^{-1}$ and 0 Gt yr$^{-1}$ in the previous version, stressing the importance of accurately modelling individual SMB components. In this dry region, underestimation of snowfall

accumulation in RACMO2.3p1 initiated a pronounced feedback decreasing SMB: active drifting snow processes erode the shallow snow cover, exposing bare ice prematurely and moving the equilibrium line too far inland (Figs. 8b and c).

### 4.3 K-transect

The K-transect in southwest Greenland consists of eight stake sites where SMB is measured annu-

ally (yellow dots in Fig. 6a) (Van de Wal et al., 2012; Machguth et al., 2016). Figure 9a compares




modelled (RACMO2.3p2 as blue dots and RACMO2.3p1 in red), with observed SMB (black dots) along the transect, averaged for the period 1991-2015. Using mean annual SMB at each station, the updated model shows a smaller bias (-30 mm w.e. yr$^{-1}$), reduced RMSE (-205 mm w.e. yr$^{-1}$), and a larger $R^2$ (0.97). In the low ablation zone ($<$ 600 m a.s.l.), the lower ice albedo increases runoff in

summer, locally reducing SMB. Decreased runoff in the upper ablation zone, i.e. between 600 and 1500 m a.s.l., increases SMB, improving the agreement at all sites except SHR. A negative bias in SMB remains at site S6 where ice albedo in summer (0.45 in July) is underestimated by up to 0.1 (Fig. 5d). Above the equilibrium line ($>$ 1500 m a.s.l.), in situ stake SMB measurements systematically underestimate climatic SMB, as they do not or only partly account for internal accumulation,

i.e. refreezing in the firn. For comparison at S10, we therefore use the difference between modelled total precipitation and melt instead of SMB, decreasing the bias and RMSE in RACMO2.3p2 by 260 mm w.e. yr$^{-1}$ and 200 mm w.e. yr$^{-1}$ to -40 mm w.e. yr$^{-1}$ and 210 mm w.e. yr$^{-1}$, respectively. Measured and modelled SMB-to-elevation gradients are estimated using a linear regression: 3.21 mm w.e. m$^{-1}$ from observations, 2.62 mm w.e. m$^{-1}$ in RACMO2.3p1, and 3.16 mm w.e. m$^{-1}$ in

RACMO2.3p2, indicating a notable improvement in model performance along the K-transect.

Figures 9b and c show time series of measured (dashed lines) and modelled SMB (continuous lines; RACMO2.3p2) at each site along the K-transect for the period 1991-2016. The model realistically captures inter-annual variability in the SMB signal although substantial biases remain at stations SHR and S6 (Table 5).

**4.4 Southeast Greenland**

Southeast Greenland experiences topographically forced precipitation maxima in winter, followed by high melt rates in summer, allowing for the formation of perennial firn aquifers (Forster et al., 2014; Koenig et al., 2014). In April 2014, an AWS was installed in the aquifer zone of the southeast GrIS (yellow dot in Fig. 6a). In August 2015, the AWS was relocated from 1563 m a.s.l (66.18°N

and 39.04°W) to 1663 m a.s.l (66.36°N and 39.31°W). Figure 10 shows time series of snow albedo and cumulative snow melt energy (expressed in mm w.e.) modelled by RACMO2.3p2 (blue lines) and RACMO2.3p1 (red lines), and calculated from the AWS data (yellow lines) for the summer of 2014. The comparison is limited to 2014 because of a 3 months data gap in summer 2015.

As melt wets the snow in summer, surface albedo gradually decreases from values typical for dry

fresh snow (0.85) to wet old snow ($\sim$0.75) in late summer, before sharply increasing again when a new fresh snow cover is deposited (yellow line in Fig. 10a). In the previous model version, surface albedo could drop to values as low as $\sim$0.66 in summer (JJA), e.g. days 152 to 243, underestimating albedo by 0.04 on average. The bias is reduced to 0.01 in RACMO2.3p2 as combined lower soot concentration and decreased grain size of refrozen snow increase the surface albedo. The remaining

small negative bias is mostly ascribable to a too rapid snow metamorphism from fresh to old snow that leads to a premature drop in surface albedo, e.g. days 140 to 160. Sporadic fresh snow deposition





over older snow, characterised by sharp peaks in surface albedo during summer, are well timed by the model. Consequently, the cumulative melt obtained at the end of summer (702 mm w.e.; blue line in Fig. 10b) is reduced by ~100 mm w.e. relative to RACMO2.3p1 (red line), a significant

improvement when compared to observations (639 mm w.e.; yellow line).

## 5   Results: SMB of the contiguous ice sheet

### 5.1   Modelled SMB at 11 km

In Figure 11, we evaluate modelled SMB in RACMO2.3p2 using 182 measurements collected in the GrIS accumulation zone (white dots in Fig. 1) and 1073 stake observations from 213 sites located in

the ablation zone (yellow dots in Fig. 1). The increased precipitation in the GrIS interior reduces the negative bias in the 11 km product (blue dots in Fig. 11a) compared to the previous model version (red dots in Fig. 11a). For the full data set, a significant bias of -22 mm w.e. $yr^{-1}$ and RMSE of 72 mm w.e. $yr^{-1}$ remain in RACMO2.3p2. Sites experiencing the highest precipitation rates on the steep slopes of southeast Greenland ($> 0.5$ m w.e. $yr^{-1}$) primarily contribute to this bias. If only

values $< 0.5$ m w.e. $yr^{-1}$ are considered (156 measurements), the bias and RMSE decrease from -26 mm w.e. $yr^{-1}$ and 52 mm w.e. $yr^{-1}$ in RACMO2.3p1 to only -7 mm w.e. $yr^{-1}$ and 49 mm w.e. $yr^{-1}$ in RACMO2.3p2. In the ablation zone (Fig. 11b), the updated model performs as well as the previous version (Noël et al., 2016) although SMB remains overestimated in the lower sectors, caused by inaccurately resolved steep slopes, low ice albedo and relatively large turbulent fluxes at

the GrIS margins, which require further downscaling (see Section 5.2).

Integrated over the GrIS, modelled SMB has increased by 66 Gt $yr^{-1}$ (415 Gt $yr^{-1}$; +19%) compared to the previous version. This difference is dominated by a significant increase in SMB in the percolation zone of the GrIS, driven by reduced meltwater runoff (61 Gt $yr^{-1}$ or -22%) and reduced sublimation (10 Gt $yr^{-1}$ or -24%), while precipitation decreased by less than 1% (5 Gt

$yr^{-1}$); the latter can be explained by the smaller GrIS area (~10,000 $km^2$ or 0.6%) in the new ice mask. We deem these changes in the 11 km fields to be realistic. For the poorly resolved marginal areas, the SMB product requires further statistical downscaling to reproduce the high melt rates in these rugged regions at the ice sheet margins. At 11 km resolution, runoff is locally underestimated by up to 6 m w.e. $yr^{-1}$, e.g. station QAS_L in southern Greenland (red stars in Fig. 11b).

### 5.2   Downscaled SMB to 1 km

To solve these issues at the margins, we apply the downscaling technique described in Noël et al. (2016), which includes elevation and ice albedo corrections. As a result, modelled runoff increases by 82 Gt $yr^{-1}$ (~37%) to 305 Gt $yr^{-1}$ for the period 1958-2015, compared to the 11 km product, and the SMB bias and RMSE in the GrIS ablation zone are reduced by 480 and 460 mm w.e. $yr^{-1}$,

respectively. The error at QAS_L is reduced to 2 m w.e. $yr^{-1}$ (red stars in Fig. 11c). A major




improvement upon Noël et al. (2016) is that no additional precipitation correction is required here as the remaining negative bias in the GrIS interior has been almost eliminated in RACMO2.3p2 (Fig. 11a). At 1 km resolution, precipitation contributes 693 Gt yr$^{-1}$ to GrIS SMB. Relative to the 11 km product, GrIS-integrated SMB at 1 km decreases by 59 Gt yr$^{-1}$ (-14%) to 356 Gt yr$^{-1}$, in line with our previous estimate of 338 Gt yr$^{-1}$ (+5%) (Noël et al., 2016). This confirms once more that 11 km resolution is insufficient to resolve runoff patterns over narrow ablation zones and small outlet glaciers, and that further downscaling is essential to obtain realistic GrIS SMB.

## 6 Remaining limitations and challenges

### 6.1 Model resolution

Extensive model evaluation confirms that RACMO2.3p2 realistically reproduces the contemporary climate and SMB of Greenland, although significant biases remain. However, while a 11 km grid is sufficient to resolve large-scale inland SMB patterns, it does not well resolve irregular, low-lying regions at the GrIS margins where runoff peaks. There, the main issue remains to accurately resolve total runoff of meltwater from narrow ablation zones and small outlet glaciers. This demonstrates the need for higher resolution (statistically or dynamically) downscaled products, e.g. the 1 km product as presented here, for regional mass balance studies.

An alternative approach is to carry out a dedicated Greenland simulation at higher spatial resolution, e.g. 5.5 km (Langen et al., 2017; Mottram et al., 2017). This increase in resolution does lead to better resolved SMB gradients over marginal glaciers, without exceeding the physics constraints of a hydrostatic model like RACMO2. Subsequently applying the statistical downscaling technique to this 5.5 km product would likely result in further improvements.

### 6.2 Turbulent fluxes

Another model limitation stems from the turbulent fluxes scheme. While LHF remains generally small and contributes little to the energy budget, accurate SHF is crucial to capture extreme melt events along the GrIS margins (Fausto et al., 2016), such as those that occurred in summer 2012 (Nghiem et al., 2012). However, SHF shows significant biases in RACMO2.3p2 in low-lying regions at the GrIS margins. Improving the representation of the GrIS surface roughness and surface elevation using higher spatial resolution could reduce these biases.

### 6.3 Surface albedo

Snow melt rate is highly sensitive to soot concentration in snow (Van Angelen et al., 2012). Although assumed to be constant in time and space in RACMO2, Takeuchi et al. (2014) show a heterogeneous distribution of impurities (soot, dust, microbiological material) over the GrIS, with a gradual increase towards lower elevations due to a) the proximity of dust sources in the tundra region and, b)



downslope transport of previously deposited soot by meltwater runoff.

Over bare ice, the accumulation of cryoconites and the growth of algae play a major role in reducing surface albedo (Musilova et al., 2016; Stibal et al., 2017). Therefore, explicitly modelling impurity concentration on ice, as described in Cook et al. (2017a,b), could substantially improve melt estimates. Future climate projections should include such a bio-darkening feedback (Tedesco et al., 2016).

**7   Conclusions**

We present a detailed evaluation of the regional climate model RACMO2.3p2 (1958-2016) over the Greenland ice sheet (GrIS). The updated model generates more inland precipitation at the expense of marginal regions, reducing the dry bias in the GrIS interior. Impurity concentration in snow, i.e. soot, has been decreased by a factor of two, minimising the melt rate overestimation in the GrIS

percolation zone. We demonstrate that the model successfully reproduces the contemporary climate of the GrIS compared to daily meteorological records and radiative energy flux measurements from 23 AWS sites. Apart from the ultimate margins, the model also proves to accurately capture the seasonal cycle of radiative and turbulent heat fluxes as well as surface albedo along the K-transect in southwest Greenland. Compared to SMB observations, RACMO2.3p2 generally improves on the

previous version, especially in the extensive GrIS interior. SMB improvements are also found along the K-transect as well as in northeast and southeast Greenland. This model version will be used for future climate scenario projections at 11 km resolution. Nonetheless, since runoff from narrow glaciers in the GrIS margins remains poorly resolved at this resolution, it is necessary to further statistically downscale present-day and future SMB fields to higher spatial resolutions for use in

regional mass balance studies.

**8   Author contribution**

B. N., W. J. B., J. M. W. and M. R. B. conceived this study, decided on the new model settings and performed the analysis and synthesis of the data sets. B. N. performed the model simulations and led the writing of the manuscript. J. T. M. L., E. M., P. K. M. and L. H. U. contributed to the development

of the model. D. A., S. L., C. J. P. P. S. and R. S. W. W. processed and provided observational data sets. All authors contributed to discussions in writing this manuscript.

**9   Data availability**

RACMO2.3p2 data at 11 km (1958-2016), and a daily downscaled product at 1 km resolution are available from the authors without conditions.



*Acknowledgements.*  B. Noël, W. J. van de Berg, J. M. van Wessem, R. S. W. van de Wal and M. R. van den
Broeke acknowledge support from the Polar Programme of the Netherlands Organization for Scientific Research
(NWO/ALW) and the Netherlands Earth System Science Centre (NESSC), as well as the European Centre for
Medium-range Weather Forecasts (ECMWF) for hosting simulations and providing computation time.





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





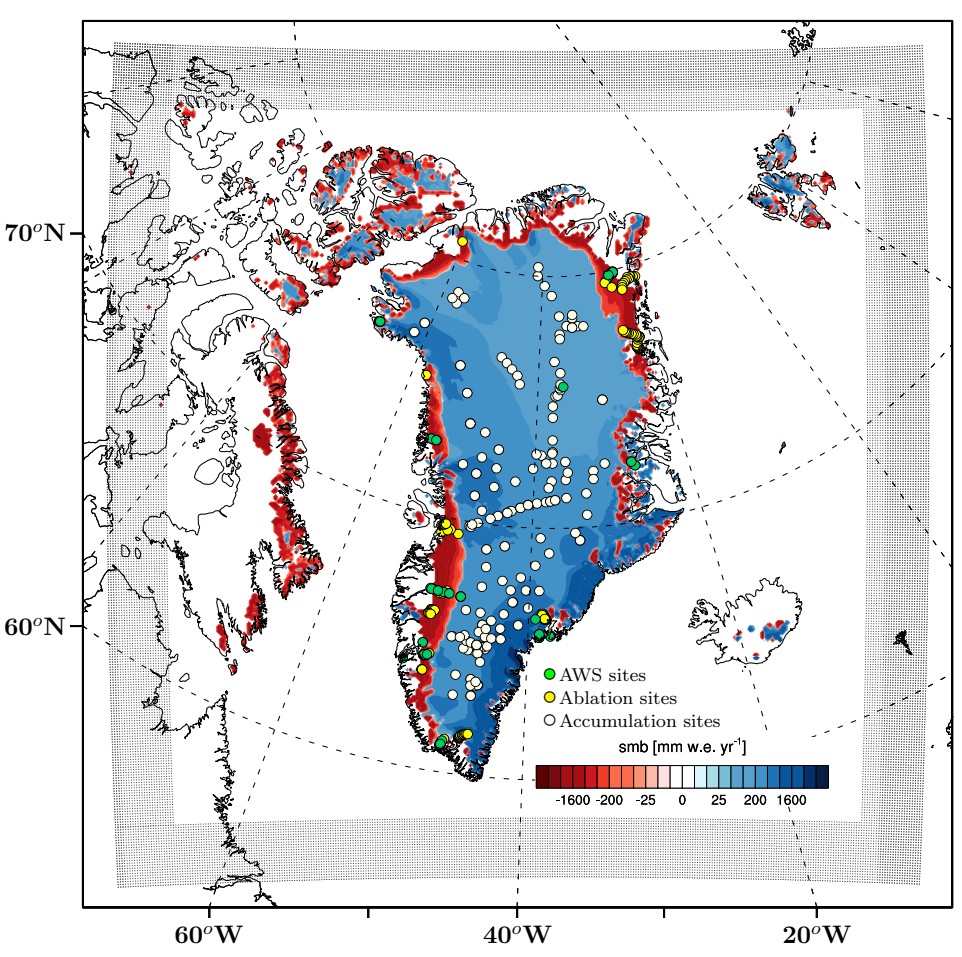

**Fig. 1.** SMB (mm w.e. yr$^{-1}$) modelled by RACMO2.3p2 at 11 km resolution for 2016. Black dots delineate the relaxation zone (24 grid cells) where the model is forced by ERA re-analyses. Ablation sites (213) are displayed as yellow dots, accumulation sites (182) as white dots, and AWS locations (23) are represented in green.



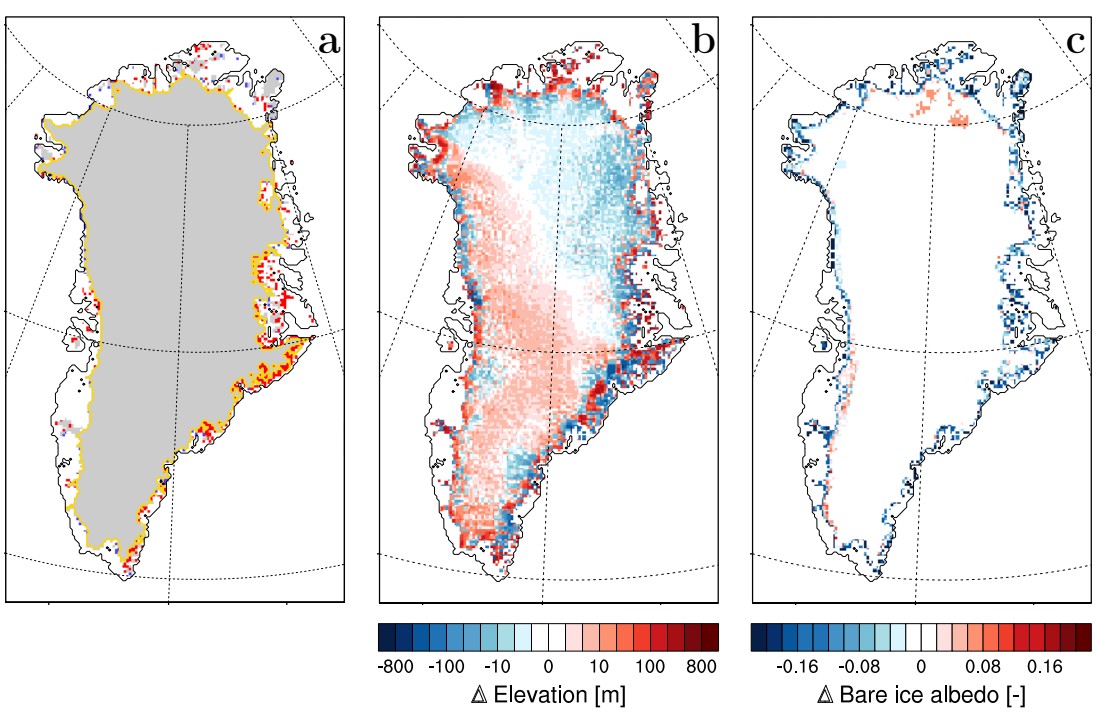

**Fig. 2.** Difference in a) ice mask b) surface elevation and c) bare ice albedo between RACMO2.3p2 and RACMO2.3p1. In Fig. 2a, the common ice mask for both model versions is displayed in grey, the ice sheet area is outlined in yellow; additional and removed ice-covered cells in RACMO2.3p2 are shown in red and blue, respectively.





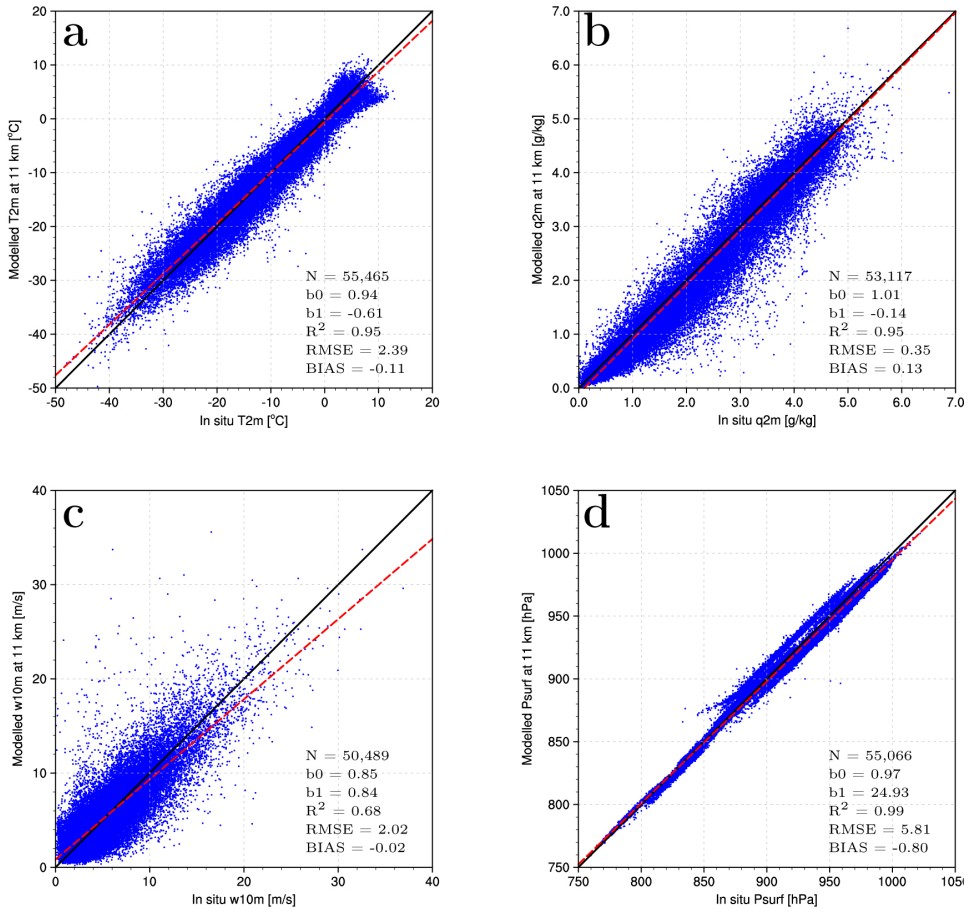

**Fig. 3.** Comparison between modelled and observed a) 2-m temperature ($T_{2m}$, °C), b) 2-m specific humidity ($q_{2m}$, g kg$^{-1}$), c) 10-m wind speed ($w_{10m}$, m s$^{-1}$) and d) surface pressure (Psurf, hPa) collected at 23 AWS (green dots in Fig. 1). For each variable, the linear regression including all records is displayed as red dashed line. Statistics including number of records (N), regression slope (b0) and intercept (b1), determination coefficient ($R^2$), bias and RMSE are listed for each variable.



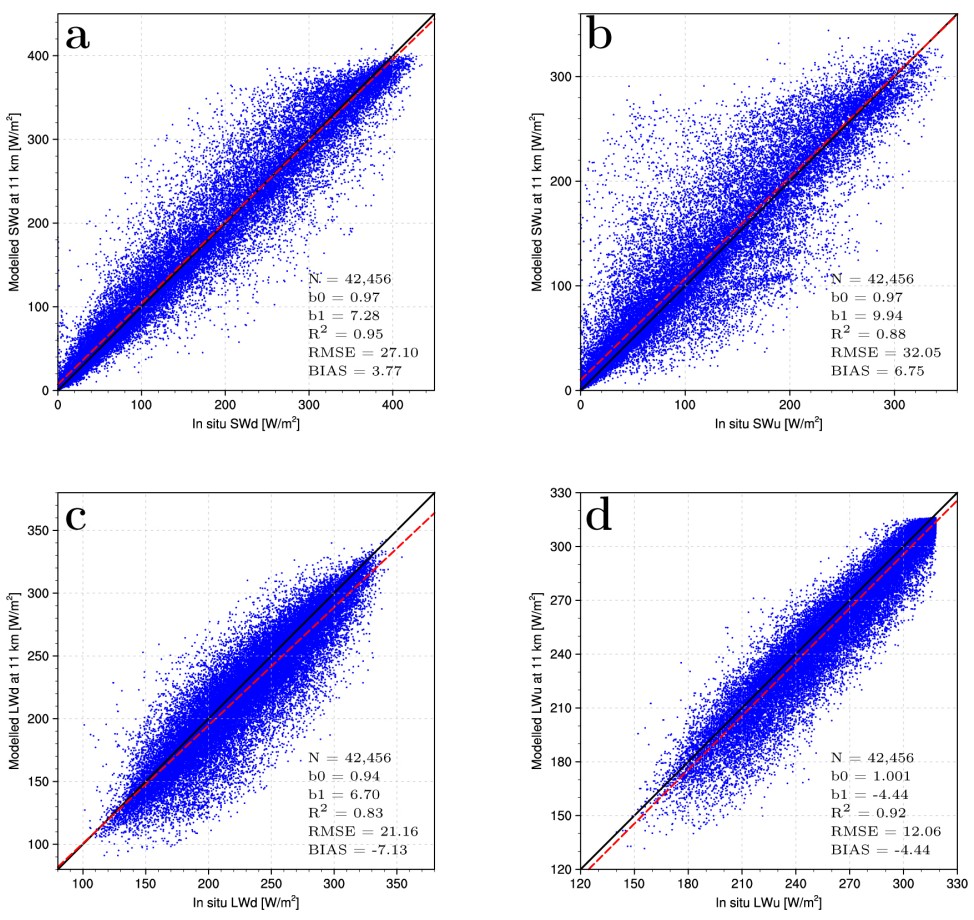

**Fig. 4.** Comparison between daily average modelled and observed a) shortwave downward, b) shortwave upward, c) longwave downward and d) longwave upward radiation (W m$^{-2}$) collected at 23 AWS (green dots in Fig. 1). For each variable, regression including all records is displayed as red dashed line. Statistics including number of records (N), the linear regression slope (b0) and intercept (b1), determination coefficient (R$^2$), bias and RMSE are listed for each variable.





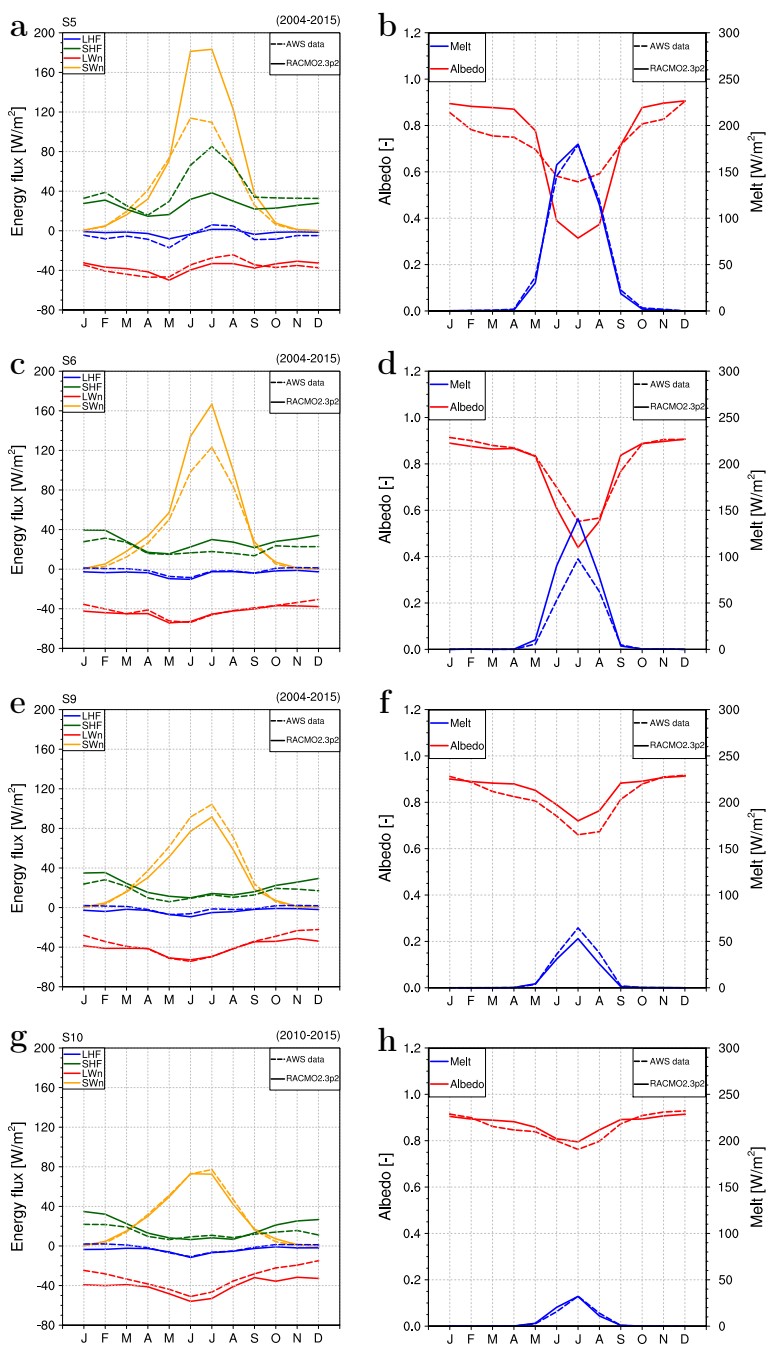

**Fig. 5.** Observed and modelled monthly mean a) turbulent and net shortwave/longwave fluxes (W m$^{-2}$) and b) surface albedo and surface melt energy (W m$^{-2}$) at site S5 for 2004-2015. Similar results are shown at S6 for 2004-2015 (c and d), S9 for 2009-2015 (e and f) and S10 for 2010-2015 (g and h).





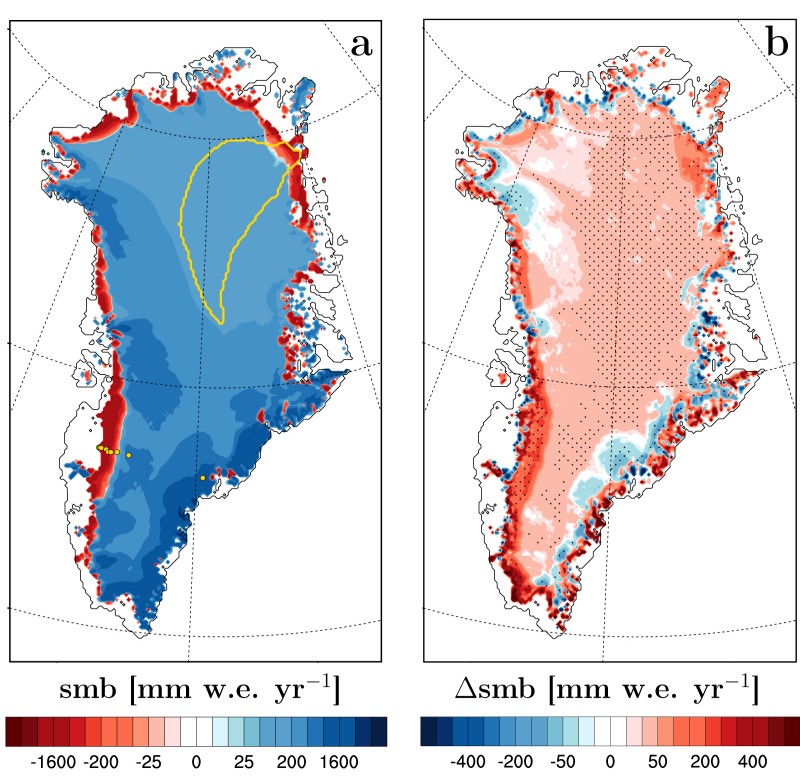

**Fig. 6.** a) SMB (mm w.e. yr$^{-1}$) averaged for the period 1958-2015. The combined Zachariae Isstrøm and Nioghalvfjerdsbrae (79N) glacier basins are delineated by the yellow line. Yellow dots locate the K-transect measurement sites in western Greenland and the single AWS operated in southeast Greenland. b) SMB difference (mm w.e. yr$^{-1}$) between RACMO2.3p2 and RACMO2.3p1 for the period 1958-2015. Areas showing significant difference are stippled in Fig. 6b: difference exceeds one unit of standard deviation of the difference between the two model versions.





**Fig. 7.** Difference in SMB components (mm w.e. yr$^{-1}$) between RACMO2.3p2 and RACMO2.3p1 averaged for the period 1958-2015. Areas showing significant difference are stippled: the difference exceeds one unit of standard deviation of the difference between the two model versions.



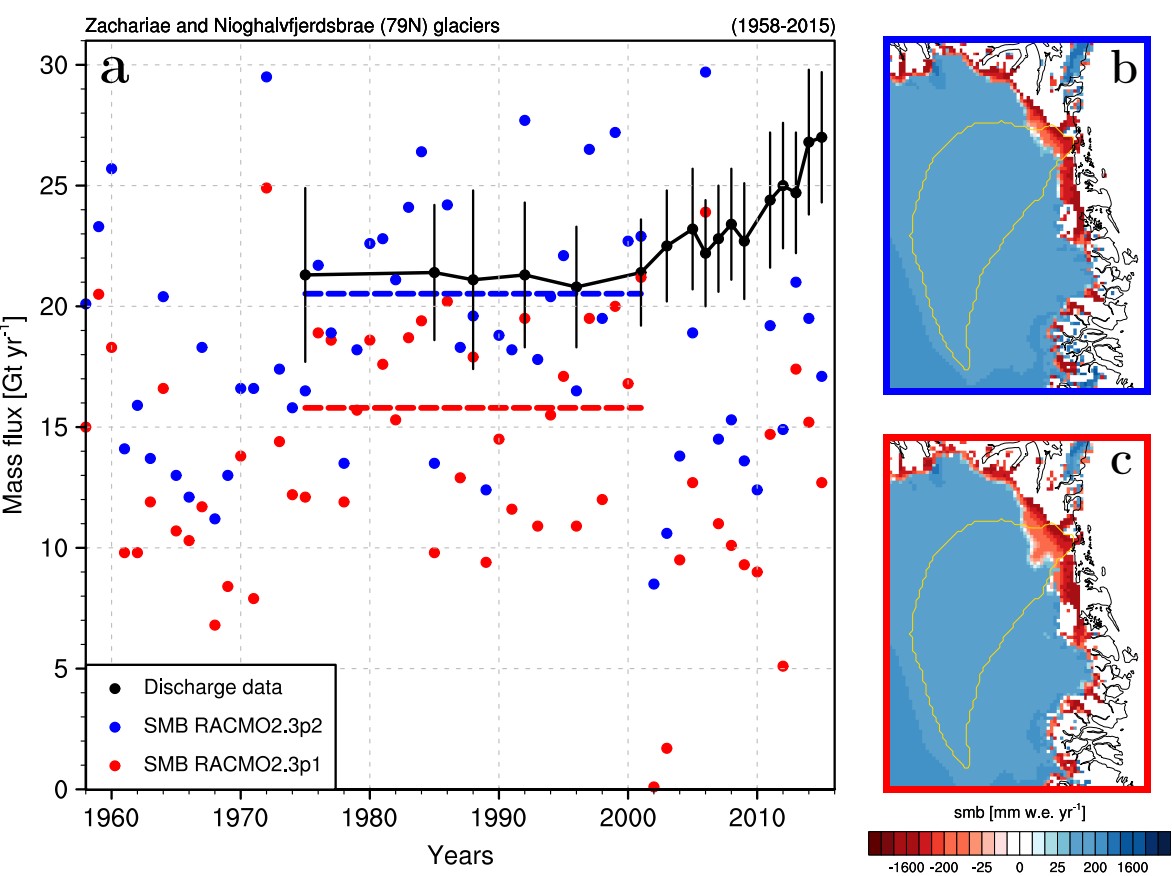

**Fig. 8.** a) Modelled basin-integrated SMB in RACMO2.3p2 (blue dots) and RACMO2.3p1 (red dots) and ice discharge estimates (black dots, Mouginot et al. (2015)) from the glacier basins of Zachariae Isstrøm and Nioghalvfjerdsbrae (79N) in northeast Greenland (yellow line in Figs. 8b and c) for the period 1975-2015. Dashed lines represent average SMB for 1975-2001. Mean SMB as modelled by b) RACMO2.3p2 and c) RACMO2.3p1 in northeast Greenland for the period 1958-2015.



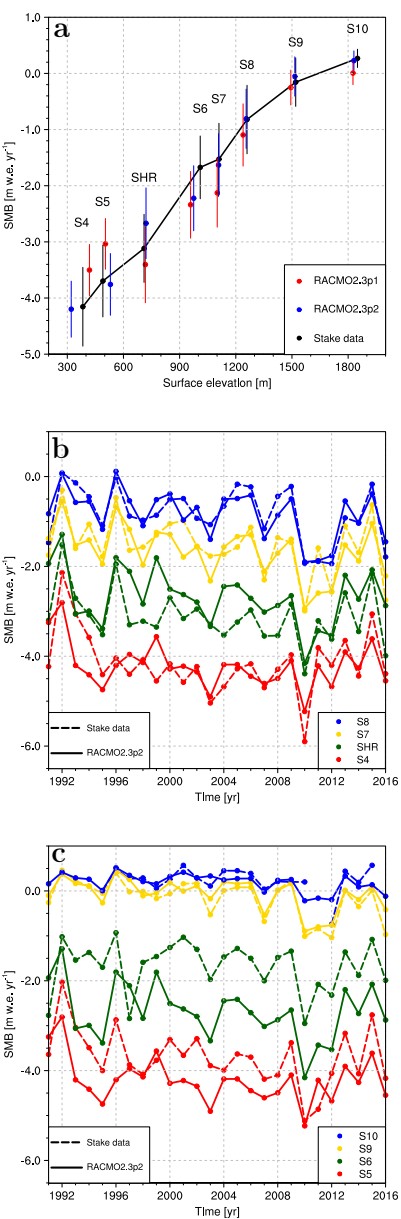

**Fig. 9.** a) Observed and simulated SMB (m w.e. yr$^{-1}$) along the K-transect in west Greenland (67°N), averaged for the period 1991-2015. The observed SMB (black dots) at S4, S5, SHR, S6, S7, S8, S9 and S10 are based on annual stake measurements; S10 observations cover 1994-2015. The coloured bars represent the standard deviation (1$\sigma$) around the 1991-2015 modelled and observed mean value. Modelled SMB at stake sites are displayed for RACMO2.3p2 (blue dots) and RACMO2.3p1 (red dots). Fig. 9b shows time series of modelled (continuous lines) and observed (dashed lines) annual SMB at stakes S4, SHR, S7 and S8 for the period 1991-2016. Similar time series are shown for the S5, S6, S9 and S10 in Fig. 9c.



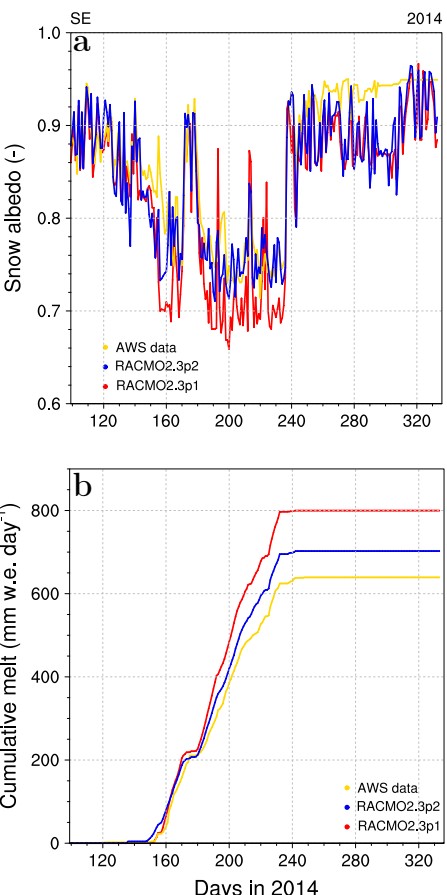

**Fig. 10.** Time series of a) daily snow albedo, b) cumulative surface melt (mm w.e. per day) modelled by RACMO2.3p2 (blue lines), RACMO2.3p1 (red lines) and measured (yellow lines) at the southeast AWS (66°N; 33°W; 1563 m a.s.l.) during summer 2014.





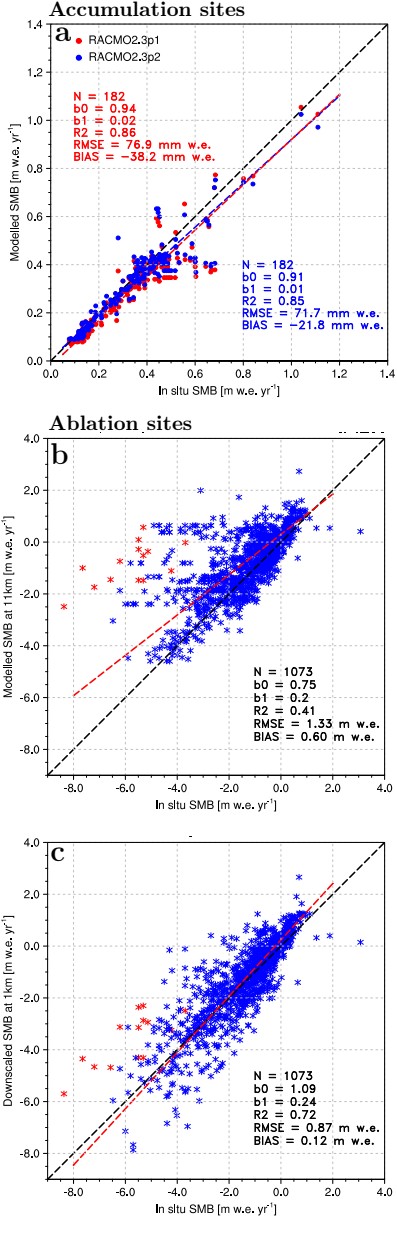

**Fig. 11.** Comparison between a) modelled, i.e. RACMO2.3p2 (blue) and RACMO2.3p1 (red) at 11 km, and observed SMB (m w.e. yr$^{-1}$) collected in the GrIS accumulation zone (white dots in Fig. 1). Regressions for RACMO2.3p2 (blue) and version 2.3 (red) are displayed as dashed lines. Comparison between SMB measurements from the GrIS ablation zone (yellow dots in Fig. 1) and b) original RACMO2.3p2 data at 11 km, c) downscaled product at 1 km. Red stars correspond to measurements collected at station QAS_L at the southern tip of Greenland. Regression including all records is displayed as red dashed line in Figs. 11b and c. Main statistics including number of records (N), regression slope (b0) and intercept (b1), determination coefficient (R$^2$), bias and RMSE are listed for each graph.





| AWS | S5 | Obs. | RACMO2.3p1 | | | RACMO2.3p2 | | |
|---|---|---|---|---|---|---|---|---|
| Variable | unit | mean | bias | RMSE | $R^2$ | bias | RMSE | $R^2$ |
| $SW_d$ | $W/m^2$ | 109.5 | 26.2 | 33.1 | 0.99 | 20.7 | 27.2 | 0.98 |
| $SW_u$ | $W/m^2$ | -70.9 | 15.8 | 25.0 | 0.93 | 4.5 | 34.3 | 0.74 |
| $LW_d$ | $W/m^2$ | 241.4 | -16.7 | 18.5 | 0.97 | -11.8 | 13.4 | 0.97 |
| $LW_u$ | $W/m^2$ | -278.3 | -13.2 | 15.5 | 0.98 | -12.1 | 14.2 | 0.98 |
| SHF | $W/m^2$ | 41.1 | -13.1 | 22.2 | 0.50 | -15.3 | 22.4 | 0.66 |
| LHF | $W/m^2$ | 5.3 | 2.6 | 5.6 | 0.72 | 3.4 | 6.5 | 0.64 |
| ME | $W/m^2$ | 42.6 | -6.8 | 18.0 | 0.96 | -0.4 | 11.9 | 0.97 |
| ALB | ( - ) | 0.74 | 0.03 | 0.09 | 0.75 | -0.004 | 0.14 | 0.72 |
| $T_{2m}$ | °C | -6.4 | -2.3 | 2.6 | 0.99 | -2.0 | 2.2 | 0.992 |

**Table 1.** Modelled and observed mean SEB components and statistics of the differences (2004-2015) at station S5 in the lower ablation zone (490 m a.s.l.). Statistics include means of measurements collected at S5, model bias (RACMO2.3pX - observations), RMSE of the bias as well as the determination coefficient of monthly mean data. Fluxes are set positive towards the surface.

| AWS | S6 | Obs. | RACMO2.3p1 | | | RACMO2.3p2 | | |
|---|---|---|---|---|---|---|---|---|
| Variable | unit | mean | bias | RMSE | $R^2$ | bias | RMSE | $R^2$ |
| $SW_d$ | $W/m^2$ | 131.6 | 9.7 | 12.7 | 0.997 | 6.0 | 9.1 | 0.997 |
| $SW_u$ | $W/m^2$ | -95.8 | -2.9 | 16.3 | 0.97 | -3.8 | 16.3 | 0.97 |
| $LW_d$ | $W/m^2$ | 222.3 | -8.8 | 11.4 | 0.96 | -2.7 | 6.5 | 0.97 |
| $LW_u$ | $W/m^2$ | -263.6 | -1.6 | 4.0 | 0.991 | -0.4 | 3.2 | 0.992 |
| SHF | $W/m^2$ | 20.8 | 9.8 | 11.4 | 0.67 | 7.0 | 8.7 | 0.70 |
| LHF | $W/m^2$ | 1.6 | -3.9 | 5.2 | 0.42 | -2.4 | 3.3 | 0.64 |
| ME | $W/m^2$ | 18.7 | 10.6 | 22.0 | 0.96 | 8.3 | 18.1 | 0.97 |
| ALB | ( - ) | 0.81 | 0.02 | 0.06 | 0.89 | -0.02 | 0.06 | 0.89 |
| $T_{2m}$ | °C | -10.9 | 0.4 | 0.8 | 0.994 | 0.7 | 1.0 | 0.995 |

**Table 2.** Modelled and observed mean SEB components and statistics of the differences (2004-2015) at station S6 in the upper ablation zone (1010 m a.s.l.). Statistics include means of measurements collected at S6, model bias (RACMO2.3pX - observations), RMSE of the bias as well as the determination coefficient of monthly mean data. Fluxes are set positive towards the surface.

| AWS | S9 | Obs. | RACMO2.3p1 | | | RACMO2.3p2 | | |
|---|---|---|---|---|---|---|---|---|
| Variable | unit | mean | bias | RMSE | $R^2$ | bias | RMSE | $R^2$ |
| $SW_d$ | $W/m^2$ | 141.2 | 2.2 | 6.6 | 0.998 | -1.5 | 7.8 | 0.998 |
| $SW_u$ | $W/m^2$ | -106.5 | 3.5 | 9.4 | 0.991 | 3.5 | 7.6 | 0.994 |
| $LW_d$ | $W/m^2$ | 217.8 | -10.1 | 14.1 | 0.93 | 3.1 | 8.9 | 0.94 |
| $LW_u$ | $W/m^2$ | -255.2 | -1.9 | 5.0 | 0.99 | 0.5 | 3.6 | 0.99 |
| SHF | $W/m^2$ | 15.8 | 7.0 | 9.2 | 0.68 | 5.2 | 7.3 | 0.74 |
| LHF | $W/m^2$ | 0.8 | -3.8 | 5.4 | 0.20 | -2.8 | 4.0 | 0.42 |
| ME | $W/m^2$ | 12.0 | -0.7 | 7.8 | 0.89 | -2.4 | 7.0 | 0.96 |
| ALB | ( - ) | 0.82 | 0.02 | 0.05 | 0.79 | 0.03 | 0.06 | 0.83 |
| $T_{2m}$ | °C | -13.3 | -0.04 | 0.7 | 0.994 | 0.5 | 0.8 | 0.996 |

**Table 3.** Modelled and observed mean SEB components and statistics of the differences (2009-2015) at station S9 close to the equilibrium line (1520 m a.s.l.). Statistics include means of measurements collected at S9, model bias (RACMO2.3pX - observations), RMSE of the bias as well as the determination coefficient of monthly mean data. Fluxes are set positive towards the surface.





| AWS | S10 | Obs. | RACMO2.3p1 | | | RACMO2.3p2 | | |
|---|---|---|---|---|---|---|---|---|
| Variable | unit | mean | bias | RMSE | $R^2$ | bias | RMSE | $R^2$ |
| $SW_d$ | $W/m^2$ | 141.5 | 1.7 | 7.0 | 0.998 | -1.1 | 9.2 | 0.996 |
| $SW_u$ | $W/m^2$ | -113.8 | -2.7 | 12.0 | 0.991 | 1.4 | 7.6 | 0.994 |
| $LW_d$ | $W/m^2$ | 220.4 | -14.4 | 17.2 | 0.93 | -6.1 | 10.6 | 0.94 |
| $LW_u$ | $W/m^2$ | -252.5 | -1.0 | 4.8 | 0.99 | 1.2 | 3.5 | 0.992 |
| SHF | $W/m^2$ | 11.9 | 7.6 | 10.8 | 0.57 | 6.6 | 8.2 | 0.79 |
| LHF | $W/m^2$ | -2.7 | -3.5 | 5.6 | 0.22 | -1.3 | 3.1 | 0.71 |
| ME | $W/m^2$ | 8.9 | 2.5 | 6.6 | 0.89 | -2.2 | 4.5 | 0.99 |
| ALB | ( - ) | 0.86 | -0.01 | 0.04 | 0.69 | 0.03 | 0.04 | 0.76 |
| $T_{2m}$ | °C | -14.6 | 0.5 | 1.0 | 0.991 | 1.0 | 1.4 | 0.994 |

**Table 4.** Modelled and observed mean SEB components and statistics of the differences (2010-2015) at station S10 in the accumulation zone (1850 m a.s.l.). Statistics include means of measurements collected at S10, model bias (RACMO2.3pX - observations), RMSE of the bias as well as the determination coefficient of monthly mean data. Fluxes are set positive towards the surface.

| Stakes | Obs. | RACMO2.3p1 | | | RACMO2.3p2 | | | Coordinates | | |
|---|---|---|---|---|---|---|---|---|---|---|
| SMB | mean | bias | RMSE | $R^2$ | bias | RMSE | $R^2$ | lon. (°W) | lat. (°N) | elev. (m a.s.l.) |
| S4 | -4.2 | 0.64 | 0.84 | 0.40 | -0.05 | 0.51 | 0.47 | -50.20 | 67.10 | 383 |
| S5 | -3.7 | 0.64 | 0.79 | 0.45 | -0.08 | 0.46 | 0.50 | -50.09 | 67.10 | 490 |
| SHR | -3.1 | -0.32 | 0.57 | 0.53 | 0.41 | 0.62 | 0.51 | -49.94 | 67.10 | 710 |
| S6 | -1.7 | -0.68 | 0.87 | 0.30 | -0.56 | 0.78 | 0.29 | -49.40 | 67.08 | 1010 |
| S7 | -1.5 | -0.65 | 0.75 | 0.64 | -0.15 | 0.37 | 0.68 | -49.15 | 66.99 | 1110 |
| S8 | -0.8 | -0.31 | 0.49 | 0.62 | -0.03 | 0.28 | 0.76 | -48.88 | 67.01 | 1260 |
| S9 | -0.2 | -0.13 | 0.21 | 0.83 | 0.07 | 0.16 | 0.88 | -48.25 | 67.05 | 1520 |
| S10 | 0.3 | -0.25 | 0.33 | 0.44 | -0.04 | 0.21 | 0.45 | -47.02 | 67.00 | 1850 |

**Table 5.** Modelled and observed mean annual SMB (m w.e. $yr^{-1}$) and statistics of the differences at S4, S5, SHR, S6, S7, S8 and S9 over 1991-2015; measurements at S10 are compared to modelled total precipitation minus melt for the period 1994-2015. Spatial coordinates of each site are listed.