# Peer review of "Modelling the climate and surface mass balance of polar ice sheets using RACMO2, Part 1: Greenland (1958-2016)"

_The Cryosphere, 2017_

## Referee Comment (RC1) · Anonymous Referee #1 · 18 Nov 2017

**Modelling the climate and surface mass balance of polar ice sheets using RACMO2, Part 1: Greenland (1958-2016)**

B. Noël, W. J. van de Berg, J. M. van Wessem, E. van Meijgaard, D. van As, J. T. M. Lenaerts, S. Lhermitte, P. Kuipers Munneke, C. J. P. P. Smeets, L. H. van Ulft, R. S. W. van de Wal, and M. R. van den Broeke

This paper presents an updated version of the polar version of the RACMO2 regional climate model (RACMO2p2), evaluated over the Greenland ice sheet against various observational datasets. Model updates include changes to the concentration of impurities assumed to be deposited onto the snowpack, the grain size of snow that has experienced meltwater refreezing, the albedo of superimposed ice, and the saltation coefficient for drifting snow. These changes generally result in an improved agreement between modeled and observed atmospheric variables, radiative fluxes and SMB. Some biases persist and can be corrected by future improvements to model physics and parameterizations and/or downscaling of model outputs.

**General Comments**

The paper is well written and the presentation is generally clear. The paper is not novel in the sense of presenting new model physics or parameterizations, but the changes to RACMO2 that are presented seem to have a significant impact on representation of the Greenland ice sheet surface and the agreement between modeled and observed SMB. A detailed validation of this updated version of the model has been conducted. The paper is therefore likely to be of interest to the cryospheric and climate modeling communities, and is important in providing details about and validation of a new version of RACMO that will be used for future studies of Greenland mass balance. I feel that the paper should be published in the Cryosphere after some relatively minor concerns are addressed below.

- 1. One general concern is the use of net surface energy balance to indicate energy available for melting. If the snow temperature is below 0°C, this energy must first be used to warm the snowpack before contributing to melting. In the ablation zone during summer where temperatures are close to freezing, most of the net energy goes to melting, but some of it must go into warming the snow/ice. The authors should revise the manuscript to refer to the net energy balance rather than melt energy, or explain why they can assume that the net energy balance can be considered melt energy.
- 2. Regarding comparisons between RACMO2.3p1 and observations for some of the plots, it would be useful to see how much RACMO2.3p2 improves on the previous version with respect to meteorological variables. Adding RACMO2.3p1 to Figs. 3, 4, and 5 could potentially make them difficult to interpret, but I think that the authors should at least provide some statistics with regard to the meteorological comparison (corresponding to statistics for Figs. 3 and 4). It would be nice to see the corresponding figures for

RACMO2.3p1 in the supplementary material as well. For RACMO2.3p1 outputs corresponding to Fig. 5, there is a similar figure in Noël et al. (2015) as the authors mention, and I think the tables provide enough information to understand the improvement.

3. In Section 3.3 and Tables 1-5, the signs of biases for upward and downward fluxes and interpretation in the text are confusing and sometimes inconsistent. It seems that the biases are generally considered with respect to the absolute value of fluxes (e.g. a negative bias for an upward flux is and underestimate of the upward flux) but this is not always the case. The calculation of net fluxes is also inconsistent. Mostly, the upward flux bias is subtracted from the downward flux bias, but not always.

The authors should make sure that the signs for all biases are consistent, and that the text interprets the direction of the biases correctly. The authors could use the same conventions for both fluxes and biases, but this should be made clear in the figure captions and in the text, to remind the reader that a positive bias for an upward flux is an underestimate of the upward flux. For example, if SWu (Obs) is -70.9, and the RACMO2.3p2 bias (RACMO2.3p2 – obs) is 4.5, the authors can indicate that the magnitude of SWu is underestimated by 4.5 in RACMO2.3p2.

Alternately the authors could define upward fluxes as being positive, which is more intuitive, and Eq. 1 could be changed so that upward fluxes are subtracted rather than added.

4. The section on the Northeast Greenland ice stream seems a bit out of place with respect to other sections. The authors should provide some more background at the beginning of Section 4.2 discussing how the ice discharge measurements can be used to evaluate SMB. The discharge measurements should also be mentioned in Section 2.5. Also, the authors should provide a rationale for their assumption of equilibrium between SMB and discharge for the northeast Greenland basin for the 1958-2015 period. The better agreement with discharge measurements suggests an improvement to SMB, but doesn't necessarily prove that SMB is accurate. This should be clarified in Section 4.2.

**Specific Comments**

- 1. **P. 1, Line 10**: Be more specific here. How are the patterns "better resolved"?
- 2. **P. 1, Line 13:** "future climate scenario projections" is unclear. Do the authors mean "projections of GrIS climate and SMB in response to future climate scenarios"?
- 3. P. 2, Line 19: Change "model simulations" to "RCM simulations"

- 4. **P. 2, Lines 25-26:** This phrase is confusing. Also, I don't believe the Box (2013) approach used data assimilation. Suggest revising "and data assimilation ... accumulation measurements..." to read: "and reconstruction of SMB obtained by combining RCM outputs with temperature and ice core accumulation measurements..."
- 5. **P. 2, Lines 27-28:** Add reference detailing CESM future simulations (Vizcaíno et al., 2014):

Vizcaíno, M., Lipscomb, W. H., Sacks, W. J., and van den Broeke M.: Greenland surface mass balance as simulated by the Community Earth System Model. Part II: Twenty-first-century changes, Journal of Climate, 27, 215-226, doi: 10.1175/JCLI-D-12-00588.1, 2014.

The authors might consider citing this study that presents a simulation of SMB from the GEOS-5 model:

Cullather, R. I., Nowicki, S. I., Zhao, B., and Suarez, M. J.: Evaluation of the surface representation of the Greenland ice sheet in a general circulation model, Journal of Climate, 27, 4835-4856, doi: 10.1175/JCLI-D-13-00635.1, 2014.

- 6. **P. 2 Line 49 P. 3 Line 1:** There are likely also improvements that could be made regardless of resolution, that a high-resolution simulation could not fix. Perhaps mention this also.
- 7. **P. 3, Line 56:** Could the authors provide a reference for RACMO2.3p1?
- 8. **P. 4, Lines 82-84:** As noted above, the net energy absorbed by the snowpack must be used to raise the surface temperature to the melting point before it can be used for melting. "M" should therefore be changed to "Enet" and this sentence should be revised accordingly.
- 9. **P. 4, Line 87:** Add "net" before "sensible and latent turbulent heat fluxes" for clarity.
- 10. **P. 5, Lines 109-111:** The corrections that have been made also can affect the ablation zone, though they probably have less of an impact there. Were similar biases found in the ablation zone previously?
- 11. **P. 5, Lines 116-118:** If possible, can the authors provide evidence that supports decreasing the size of refrozen snow grains?
- 12. **P. 5, Lines 136-137:** Can the authors be a bit more specific about the levels or height at which upper atmosphere nudging is applied?
- 13. **P. 5, Lines 139-141:** Provide a few more details about this. What are the "best" profiles and how are they derived?
- 14. **P. 6, 149-151:** Are fractional areas of ice vs. tundra allowed in a RACMO grid box? If so, it would be useful to have this information here.
- 15. **P. 6, Line 156:** Please specify the version number. Is this version 5 or version 6?
- 16. **P. 6, Lines 157-160:** These sentences are a bit unclear. I think the authors are saying that MODIS values for bare ice albedo below 0.3 are replaced by a value of 0.3, and MODIS values above 0.55 are replaced with 0.55. Any grid

cells without a valid MODIS estimate are assigned a value of 0.55. Please clarify.

- 17. **P. 7, Lines 201-202:** Are these biases statistically significant? It might be useful for the reader to have this information.
- 18. **P. 7, Lines 209-210:** Can the authors be sure that the LWd underestimation leads to the LWu underestimation?
- 19. **P. 8, Line 214:** Can the authors elaborate here? Is there a difference because of heterogeneity in fresh snow distribution leading to differences between the model estimate and local measurements?
- 20. **P. 8, Lines 229-231:** It's a bit unclear that the values in parentheses are biases and not absolute magnitude of the quantities. Clarify here and where applicable elsewhere in the text, e.g. "... between overestimated SWn (bias of 16.2 W m-2)".
- 21. **P. 8, Line 231:** Make clear whether SWu is over- or underestimated. I believe it's underestimated. (See general comments.)
- 22. **P. 8, Lines 236-237:** I believe the newest MCD43A3 product includes a correction for sensor deterioration, but if v5 is used here, this still applies. Perhaps clarify with "underestimated surface albedo for the MCD43A3 v5 product"
- 23. **P. 8, Lines 238-239:** Again this is confusing because of sign conventions. If the signs of the biases follow the conventions, the net bias should be -23.9 W m-2 and not 0.3 W m-2.
- 24. P. 9, Line 252: Also add reference to Table 2 here.
- 25. **P. 9, Line 253:** Clarify under- vs. overestimated, use "~4 W m-2" to indicate that the value is approximate.
- 26. **P. 9, Lines 269-270:** According to most of the previous calculations of net flux, these terms don't compensate. There is underestimated downward flux and overestimated upward flux, so the next flux is underestimated.
- 27. P. 9, Line 271: Add reference to Table 4.
- 28. **P. 9, Lines 272-273:** Again, here the biases have been added rather than subtracted to get the net flux, in contrast with calculations for other sections.
- 29. **P. 10, Line 289:** Here the SWu bias, shown as positive in Table 1, is referred to as negative, which would make sense if conventions are followed everywhere, but is not consistent with earlier discussion (e.g. p. 9, line 264, where a positive bias for SWu is referred to as an overestimation).
- 30. **P. 11, Line 316:** The increase in refreezing is attributed to an increase in precipitation, but along the west coast, there is a decrease in precipitation in some areas. Perhaps another factor could be persistence of snow cover as a result of reduced melting.
- 31. **P. 12, Lines 353-354:** Show numbers for both model versions for comparison.
- 32. P. 12, Lines 358-362: Is this correction applied to the values in Fig. 9?
- **33. P. 13, Lines 402-405:** Provide some numbers to illustrate that the new version performs as well as the previous version.
- 34. **P. 13, Lines 419-420:** What are the new values for RMSE, bias, and error at QAS\_L?

- 35. **Tables 1-5:** The term ME is used here for melt energy, but the term M is used in the text. These should be consistent. As noted in the general comments I believe this should really refer to the net energy balance. Captions for Tables 2-5 can be reduced to "Same as Table 1 for Station..."
- 36. **Figure 11:** The red points in (a) indicate something different from (b) and (c). I feel that the authors should include RACMO2.3p2 for (a), (b) and (c), and use the same color scheme. A different color could be used to show the measurements from QAS\_L. Units for statistics should be the same for all figures if possible, and should correspond to the units in the text. Also, I believe the third line of caption: "version 2.3 (red)" should read "version 2.3p1 (red)".

**Technical Corrections**

- 1. **P. 1, Line 10:** Change "than the previous model version" to "compared with the previous model version"
- 2. P. 2, Line 29: Change "to explicitly resolve" to "of explicitly resolving"
- 3. P. 2, Line 33: Change "evaluate and improve" to "evaluating and improving"
- 4. **P. 2, Line 45:**". If the authors are still referring to previous versions of the model, change to "is underestimated" to "was underestimated".
- 5. P. 2, Line 50: Change "near-kilometre" to "near-kilometre-scale"
- 6. **P. 3, Line 55:** Change "all over Greenland." to "across the GrIS."
- 7. P. 5, Line 132: Add "an" before "11 km horizontal..."
- **8. P. 5, Lines 134-135:** Mention that the model domain is shown in Fig. 1 to make clear what Fig. 1 is showing.
- 9. P. 7, Line 191: Change "of 23 AWS" to "from 23 AWS"
- 10. P. 7, Line 192: Change "output" to "outputs".
- **11. P. 7, Line 194:** Add "and" after "10-m wind speed,"
- **12. P. 7, Line 201:** change "with a small negative bias" to "with the model exhibiting a small negative bias" for clarity.
- **13.** P. 8, Line 231 and Line 239: Change "too low cloud cover" to "underestimated cloud cover"
- **14. P. 8, Line 241:** The van den Broeke (2008) reference seems to be missing from the reference list.
- 15. P. 9, Line 258: Change "too large SHF" to "SHF to be overestimated"
- 16. P. 10, Lines 295 298: The language could be improved here. Suggested revision: "In Section 3, we discussed the overall good ability of RACMO2.3p2 to reproduce the contemporary climate of the GrIS, which is essential for estimating realistic SMB patterns. Here we compare SMB from RACMO2.3p2 and RACMO2.3p1 over the GrIS. For further evaluation, we focus on three regions where there are large differences in SMB between the two versions."
- **17. P. 11, Line 318:** Change "very GrIS margins" to "extreme margins of the GrIS"
- **18. P. 12, Lines 360-362:** This sentence is a bit wordy... suggest changing "decreasing the bias..." to "decreasing the bias by 260 mm w. e. yr-1 to -40 mm w. e. yr-1 and the RMSE by 200 mm w. e. yr-1 to 210 mm w. e. yr-1."

- 19. P. 12, Line 378: Change "3 months" to "3 month"
- 20. P. 14, Line 426: Add "an" before "11 km resolution"
- **21. P. 14, Line 434:** change "narrow ablation zones" to "the narrow ablation zone".
- 22. P. 14, Line 444: Change "to capture" to "to capturing".
- 23. P. 15, Line 455: Change "cryoconites" to "cryoconite".
- **24. P. 15, Line 467:** Change "proves to accurately capture" to "accurately captures"
- 25. **Figure 2:** Although it is not necessary since the caption provides a description, a legend on Fig. 2a would be useful for the reader.
- 26. **Figure 5:** The dashed lines on the legend are hard to distinguish from solid lines.
- 27. **Figure 9:** The black and blue colors are a bit hard to distinguish. Can the blue color be made slightly brighter? In caption, remove "the" in "for the S5"
- **28. Figure 10:** The yellow line is difficult to see. The color could be made slightly darker. Add "and" after "a) daily snow albedo"
- **29.** Fix references to follow format for *The Cryosphere*

---

## Referee Comment (RC2) · Anonymous Referee #2 · 14 Dec 2017

This is a very nice well-written and detailed paper describing the revised RACMO2.3p2 model and the effect of improved model tuning on the Greenland ice sheet surface mass balance estimates. This kind of paper is extremely important for users of SMB data to read and digest in order to understand the likely biases and uncertainties within model output and the thorough analysis, while not really presenting much novel scientific research is an important addition to the scientific canon. It is very well structured and easy to read and the authors are to be congratulated on a thorough overview. That said I have some issues, which I feel should be addressed before final publication. Points for consideration

1. In section 2.3 Model Updates, it is noted that there have been some changes to the cloud scheme but these are not discussed in much detail and it is not clear how the large adjustments to the llcrit in mixed phase and ice clouds were reached. This is a pretty serious adjustment of the model as many of the other model parameterisations in the radiation scheme are likely to be tuned to these kind of values, possibly giving erroneous results or different compensating biases. However little detail is given as to how or why the particular values for these adjustments were chosen. Nor are the effects of this adjustment alone described in any detail – for instance on lines 398 to 400 reference is made to a precipitation bias in the SE compared to observations, but it is not clear if this bias is reduced or increased from RACMO2.3p1 and if this is a result of the cloud scheme changes or for example the change in topography caused by moving to the GIMP DEM. More information on how this change in cloud parameterisations has altered precipitation in particular would be helpful, as the differences in the topography shown in figure 2 seem also to be related to the change in distribution of precipitation, at least in some locations as shown in figure 7.

2. On a similar theme, I note that the small improvement in LWd and SWd on the K-transect is reasonably attributed to the change in the cloud scheme. It would be very interesting to see if this improvement is consistent across Greenland at stations other than the K-transect. There is some reason to believe that western Greenland is often modelled well but in other regions RCMs do a less good job of reproducing observed climate variables, possibly die to biases in cloud schemes as well as the complex topography in other areas. As there is now a fairly large amount of data available from Promice stations it would be nice to see some geographical spread in the figures presented in figure 5 and tables 1-5, perhaps limited to maybe 3-4 extra stations in north, south and east Greenland to determine if the positive results from western Greenlad are replicated elsewhere.

3. Upper atmosphere relaxation is mentioned on line 136-7 but no details are given. I would like to see this expanded with details on which fields are nudged and at which

levels in the atmosphere as this is important for interpreting the atmospheric model output.

4. The authors acknowledge that boundary forcing is important for results (line 20) but the differences between results from ERA40 and ERA-Interim forced years are not explored at all. It would be helpful to have a time series of SMB and the components for the full 1958-2015 averaged over the full ice sheet for the full period. This would show if, for example, there is a marked change in precipitation or melt potentially resulting from the switch in boundary forcing in 1979 is visible in more detail. It would also give a better sense of the interannual and decadal scale variability in SMB of Greenland. Plotting these with model version p1masked with the same ice mask would also allow us to assess the differences in SMB over the full ice sheet that result from the improvements introduced here.

5. I am not quite clear if the improvements to the snow module are part of the online RACMO model or the offline firn model – I assume the former, but please clarify this in sections 2.1 and 2.3

6. On line 216 you note that AWS data is sometimes spurious, Ryan et al 2017 in GRL also showed that the siting of stations (for very good reasons!) also leads to spurious underestimation of albedo – this should probably be referenced.

---

## Author Comment (AC1) · 11 Jan 2018

Please, find attached our response to the reviewers, revised manuscript and Supplementary Material.

Please also note the supplement to this comment:
https://www.the-cryosphere-discuss.net/tc-2017-201/tc-2017-201-AC1-supplement.zip

---

## Author Response (AR1)

**Response to reviewers:**

Dear reviewers, we would like to thank you for your constructive comments on our manuscript. We tried to address most of the concerns you raised, and to apply corrections where appropriate to improve our manuscript.

To facilitate readability, our responses to reviewers are displayed in blue and modifications in the manuscript are highlighted in red. These suggested changes, together with additional minor corrections, are also displayed in red in the attached revised manuscript.

**Reviewer #1:**

This paper presents an updated version of the polar version of the RACMO2 regional climate model (RACMO2p2), evaluated over the Greenland ice sheet against various observational datasets. Model updates include changes to the concentration of impurities assumed to be deposited onto the snowpack, the grain size of snow that has experienced meltwater refreezing, the albedo of superimposed ice, and the saltation coefficient for drifting snow. These changes generally result in an improved agreement between modeled and observed atmospheric variables, radiative fluxes and SMB. Some biases persist and can be corrected by future improvements to model physics and parameterizations and/or downscaling of model outputs.

**General Comments**

The paper is well written and the presentation is generally clear. The paper is not novel in the sense of presenting new model physics or parameterizations, but the changes to RACMO2 that are presented seem to have a significant impact on representation of the Greenland ice sheet surface and the agreement between modeled and observed SMB. A detailed validation of this updated version of the model has been conducted. The paper is therefore likely to be of interest to the cryospheric and climate modeling communities, and is important in providing details about and validation of a new version of RACMO that will be used for future studies of Greenland mass balance. I feel that the paper should be published in the Cryosphere after some relatively minor concerns are addressed below.

   One general concern is the use of net surface energy balance to indicate energy available for melting. If the snow temperature is below 0°C, this energy must first be used to warm the snowpack before contributing to melting. In the ablation zone during summer where temperatures are close to freezing, most of the net energy goes to melting, but some of it must go into warming the snow/ice. The authors should revise the manuscript to refer to the net energy balance rather than melt energy, or explain why they can assume that the net energy balance can be considered melt energy.
In RACMO2, the skin temperature (Tskin) of snow and ice is derived by closing the surface energy budget (SEB), using the linearized dependencies of all fluxes to Tskin and further assuming, as a first approximate, that no melt occurs at the surface (M = 0). If the obtained Tskin exceeds the melting point, Tskin is set to 0ºC; all fluxes are then recalculated and the melt energy flux (M) is estimated by closing the SEB using Eq. 1. Therefore, snow melt is possible even though the upper snow layer remains below the melting point. In that case, the generated meltwater refreezes directly in this upper snow layer, rapidly warming it to 0ºC. This is now clarified in the manuscript: "In RACMO2, the skin temperature (Tskin) of snow and ice is derived by closing the surface energy budget (SEB), using the linearized dependencies of all fluxes to Tskin and further assuming, as a first approximate, that no melt occurs at the surface (M = 0). If the obtained Tskin exceeds the melting point, Tskin is set to 0ºC; all fluxes are then recalculated and the melt energy flux (M > 0) is estimated by closing the SEB in Eq. 1., assuming that no solar radiation can directly penetrate the snow or ice interface." As the term "melt energy" is commonly used in multiple publications, e.g. Van den Broeke et al. (2017), Van Wessem (2013, 2017), Ettema et al. (2010), and to remain consistent with previously published RACMO2 papers, we decided to keep M to refer to melt energy.

Regarding comparisons between RACMO2.3p1 and observations for some of the plots, it would be useful to see how much RACMO2.3p2 improves on the previous version with respect to meteorological variables. Adding RACMO2.3p1 to Figs. 3, 4, and 5 could potentially make them difficult to interpret, but I think that the authors should at least provide some statistics with regard to the meteorological comparison (corresponding to statistics for Figs. 3 and 4). It would be nice to see the corresponding figures for RACMO2.3p1 in the supplementary material as well. For RACMO2.3p1 outputs corresponding to Fig. 5, there is a similar figure in Noël et al. (2015) as the authors mention, and I think the tables provide enough information to understand the improvement.

As RACMO2.3p1 only covers the period 1958-2015, a direct comparison with RACMO2.3p2 (including 2016) cannot be conducted without discarding a substantial amount of data. Therefore, we decided to keep Figs. 3 and 4 as are and included similar figures for RACMO2.3p1 (1958-2015) in a Supplementary Material (see Figs. S5 and S10). These figures also list statistics for the radiative fluxes and meteorological data. We included a new table (Table 1) summarizing the main statistics in the revised manuscript together with the following sentences in Section 3.1 and 3.2, respectively. "Table 1 and Fig. S5 compare the agreement of RACMO2.3p2 and version 2.3p1 with in situ measurements. We find an overall improvement in the updated model version, showing a smaller bias and RMSE as well as an increased variance explained. Notably, the remaining negative bias in 2-m temperature (Fig. S5a) and the systematic dry bias (Fig. S5b) in RACMO2.3p1 have almost vanished in the updated model version (Figs. 3a and b)."

"Compared to the previous model version (Table 1), changes in the cloud scheme have significantly improved the representation of LWd (Figs. 4c and S10c), showing a reduced negative bias and RMSE. These modifications have also somewhat decreased the positive bias in SWd (Fig. 4a), relative to RACMO2.3p1 (Fig. S10a). In addition, LWu is notably improved in RACMO2.3p2: the remaining negative bias in LWu has almost vanished (Figs. 4d and S10d). This can be partly explained by the much better resolved 2-m temperature in RACMO2.3p2."

In Section 3.3 and Tables 1-5, the signs of biases for upward and downward fluxes and interpretation in the text are confusing and sometimes inconsistent. It seems that the biases are generally considered with respect to the absolute value of fluxes (e.g. a negative bias for an upward flux is and underestimate of the upward flux) but this is not always the case. The calculation of net fluxes is also inconsistent. Mostly, the upward flux bias is subtracted from the downward flux bias, but not always. The authors should make sure that the signs for all biases are consistent, and that the text interprets the direction of the biases correctly. The authors could use the same conventions for both fluxes and biases, but this should be made clear in the figure captions and in the text, to remind the reader that a positive bias for an upward flux is an underestimate of the upward flux. For example, if SWu (Obs) is -70.9, and the RACMO2.3p2 bias (RACMO2.3p2 − obs) is 4.5, the authors can indicate that the magnitude of SWu is underestimated by 4.5 in RACMO2.3p2. Alternately the authors could define upward fluxes as being positive, which is more intuitive, and Eq. 1 could be changed so that upward fluxes are subtracted rather than added.

As suggested, we decided to adopt the convention that all SEB fluxes are set positive. We corrected equation 1, values in the main text and in the tables accordingly. We also noticed a small bug in the script calculating statistics at site S10; this has been corrected in the text and associated table. We also made sure that LWn and SWn biases are correctly defined, and clarified when necessary if fluxes are under-overestimated.

The section on the Northeast Greenland ice stream seems a bit out of place with respect to other sections. The authors should provide some more background at the beginning of Section 4.2 discussing how the ice discharge measurements can be used to evaluate SMB. The discharge measurements should also be mentioned in Section 2.5. Also, the authors should provide a rationale for their assumption of equilibrium between SMB and discharge for the northeast Greenland basin for the 1958-2015 period. The better agreement with discharge measurements suggests an improvement to SMB, but doesn't necessarily prove that SMB is accurate. This should be clarified in Section 4.2.

We clarified as follows, in Section 2.5: "In addition, we compare modelled SMB with annual glacial ice discharge (D) retrieved from the combined Zachariae Isstrøm and Nioghalvfjerdsbrae glacier catchments in northeast Greenland (1975-2015; yellow line in Fig. 6a), presented in Mouginot et al. (2015).".

In Section 4.2: "For northeast Greenland's two main glaciers, Zachariae Isstrøm and Nioghalvfjerdsbrae (79N glacier; yellow line in Fig. 6a), solid ice discharge (D) estimates are available for the period 1975-2015 (Mouginot et al., 2015). Assuming that this glacier catchment draining ~12% of the GrIS area remained in approximate balance until ~2000 (Mouginot et al., 2015), i.e. D = SMB, measurements of D at the grounding line of these marine-terminating glaciers can be used to evaluate modelled SMB.".

Followed by: "However, it is important to note that, while good agreement is obtained between averaged SMB and D before 2001, suggesting a glacier catchment in approximate balance as in Mouginot et al. (2015), this does not necessarily confirm that spatial and temporal variability of northeast Greenland SMB is accurately resolved by the model.".

**Specific Comments**

P. 1, Line 10: Be more specific here. How are the patterns "better resolved"?
We reformulated as: "[…] better resolves the spatial patterns and temporal variability of SMB compared with the previous […]".
P. 1, Line 13: "future climate scenario projections" is unclear. Do the authors mean "projections of GrIS climate and SMB in response to future climate scenarios"?
We reformulated accordingly. Thank you.
P. 2, Line 19: Change "model simulations" to "RCM simulations"
Thank you.
P. 2, Lines 25-26: This phrase is confusing. Also, I don't believe the Box (2013) approach used data assimilation. Suggest revising "and data assimilation … accumulation measurements…" to read: "and reconstruction of SMB obtained by combining RCM outputs with temperature and ice core accumulation measurements…"
We reformulated accordingly.
P. 2, Lines 27-28: Add reference detailing CESM future simulations (Vizcaíno et al., 2014): Vizcaíno, M., Lipscomb, W. H., Sacks, W. J., and van den Broeke M.: Greenland surface mass balance as simulated by the Community Earth System Model. Part II: Twenty-first-century changes, Journal of Climate, 27, 215-226, doi: 10.1175/JCLI-D-12-00588.1, 2014. The authors might consider citing this study that presents a simulation of SMB from the GEOS-5 model: Cullather, R. I., Nowicki, S. I., Zhao, B., and Suarez, M. J.: Evaluation of the surface representation of the Greenland ice sheet in a general circulation model, Journal of Climate, 27, 4835-4856, doi: 10.1175/JCLI-D-13-00635.1, 2014.
Thank you, reformulated as follows:"Vizcaìno et al. (2013, 2014) and Cullather et al. (2014) respectively used the Community Earth System Model (CESM) at 1º resolution (~100 km) and the Goddard Earth Observing System model version 5 (GEOS-5) at 0.5º resolution (~50 km) to estimate recent and future mass losses of the GrIS."
P. 2 Line 49 – P. 3 Line 1: There are likely also improvements that could be made regardless of resolution, that a high-resolution simulation could not fix. Perhaps mention this also.
To clarify this, we inserted the following sentence: "Important modelling challenges and limitations still need to be addressed in RACMO2 regardless of the spatial resolution used: e.g. cloud representation (Van Tricht et al., 2016), surface albedo and turbulent heat fluxes (Section 6)."
P. 3, Line 56: Could the authors provide a reference for RACMO2.3p1?
We added a reference to Noël et al. (2015).
P. 4, Lines 82-84: As noted above, the net energy absorbed by the snowpack must be used to raise the surface temperature to the melting point before it can be used for melting. "M" should therefore be changed to "Enet" and this sentence should be revised accordingly.
See our previous response to General comments.
P. 4, Line 87: Add "net" before "sensible and latent turbulent heat fluxes" for clarity. Done.

P. 5, Lines 109-111: The corrections that have been made also can affect the ablation zone, though they probably have less of an impact there. Were similar biases found in the ablation zone previously?
From Fig. 9 in Noël et al. (2015) (see below), you can clearly see that compared to observations along the K-transect (black dots) SMB in RACMO2.3p1 (blue) is overestimated in the ablation zone, due to too low melt (ablation) rates.

[Figure]

P. 5, Lines 116-118: If possible, can the authors provide evidence that supports decreasing the size of refrozen snow grains?
We could not find a proper reference that validates our assumption. However, Kuipers Munneke et al. (2011) suggest that the grain size of refrozen snow should be ascribed a value larger or equal to 1000 µm. We therefore chose to set the grain size of refrozen snow to 1000 µm: this value minimizes the positive melt bias in the model across the GrIS percolation zone.

P. 5, Lines 136-137: Can the authors be a bit more specific about the levels or height at which upper atmosphere nudging is applied?
See our response to Reviewer #2.

P. 5, Lines 139-141: Provide a few more details about this. What are the "best" profiles and how are they derived?
We clarified this by reformulating as: "The model has about 40 active […] 1957 using estimates of temperature and density profiles derived from the offline […] (Ligtenberg et al., 2011). These profiles are obtained by repeatedly running IMAU-FDM over 1960-1979 forced by the outputs of the previous RACMO2.3p1 climate simulation until the firn column reaches an equilibrium."

P. 6, 149-151: Are fractional areas of ice vs. tundra allowed in a RACMO grid box? If so, it would be useful to have this information here.
To clarify this, we inserted the following sentence: "In RACMO2, a grid-cell with an ice fraction ≥ 0.5 is considered fully ice-covered.".

P. 6, Line 156: Please specify the version number. Is this version 5 or version 6?

We now mention version 5 in the revised manuscript.

P. 6, Lines 157-160: These sentences are a bit unclear. I think the authors are saying that MODIS values for bare ice albedo below 0.3 are replaced by a value of 0.3, and MODIS values above 0.55 are replaced with 0.55. Any grid cells without a valid MODIS estimate are assigned a value of 0.55. Please clarify.

Your interpretation is right, except for non-valid MODIS estimates which are set to 0.47. For clarity, we reformulated as: "In RACMO2, minimum ice albedo is set to 0.30 for […] , and a maximum value of 0.55 […].".

P. 7, Lines 201-202: Are these biases statistically significant? It might be useful for the reader to have this information.

We calculated the significance of the biases in Figs. 3-4 by comparing the mean bias listed in each scatter plot with 2 standard deviations of the biases (95% interval). None of the biases were larger than 2 standard deviations, so that these biases are determined as insignificant. As the biases are relatively small and negligible compared to the absolute value of the meteorological data and radiative fluxes, we deem that this is not necessary to mention in the manuscript.

P. 7, Lines 209-210: Can the authors be sure that the LWd underestimation leads to the LWu underestimation?

We reformulated as: "The negative biases in LWd and 2-m temperature partly lead to LWu underestimation of […].".

P. 8, Line 214: Can the authors elaborate here? Is there a difference because of heterogeneity in fresh snow distribution leading to differences between the model estimate and local measurements?

We elaborated as follows: "In RACMO2, precipitation falls vertically, i.e. no horizontal transport is allowed, and is assumed to be instantly deposited at the surface. Consequently, the spatial distribution of summer snow patches may be locally inaccurate, resulting in large albedo discrepancies when compared to point albedo measurements.".

P. 8, Lines 229-231: It's a bit unclear that the values in parentheses are biases and not absolute magnitude of the quantities. Clarify here and where applicable elsewhere in the text, e.g. "… between overestimated SWn (bias of 16.2 W m-2)".

Done.

P. 8, Line 231: Make clear whether SWu is over- or underestimated. I believe it's underestimated. (See general comments.)

Done.

P. 8, Lines 236-237: I believe the newest MCD43A3 product includes a correction for sensor deterioration, but if v5 is used here, this still applies. Perhaps clarify with "underestimated surface albedo for the MCD43A3 v5 product".

We reformulated accordingly.

P. 8, Lines 238-239: Again this is confusing because of sign conventions. If the signs of the biases follow the conventions, the net bias should be -23.9 W m-2 and not 0.3 W m-2.

This is now corrected. See also our response to General comments.

P. 9, Line 252: Also add reference to Table 2 here.

Done.

P. 9, Line 253: Clarify under- vs. overestimated, use "~4 W m-2" to indicate that the value is approximate.

Done. See also our response to General comments.

P. 9, Lines 269-270: According to most of the previous calculations of net flux, these terms don't compensate. There is underestimated downward flux and overestimated upward flux, so the next flux is underestimated.

Both Swd and SWu are underestimated. Thank you.

P. 9, Line 271: Add reference to Table 4.

Done.

P. 9, Lines 272-273: Again, here the biases have been added rather than subtracted to get the net flux, in contrast with calculations for other sections.

This is now corrected.

P. 10, Line 289: Here the SWu bias, shown as positive in Table 1, is referred to as negative, which would make sense if conventions are followed everywhere, but is not consistent with earlier discussion (e.g. p. 9, line 264, where a positive bias for SWu is referred to as an overestimation).

At line 289, we mention site S6 showing a SWu underestimate of 3.8 W/m2, while line 264 refers to site S9 close to the equilibrium line where SWu is overestimated by 3.5 W/m2. We clarified this by replacing "Here, […]" by "At site S6, […]".

P. 11, Line 316: The increase in refreezing is attributed to an increase in precipitation, but along the west coast, there is a decrease in precipitation in some areas. Perhaps another factor could be persistence of snow cover as a result of reduced melting.

Thank you. We reformulated as follows: "[…] a result of combined enhanced precipitation and reduced summer melt (delaying the disappearance of the seasonal snow cover), that increased […] (Fig. 7f).".

P. 12, Lines 353-354: Show numbers for both model versions for comparison.

The numbers previously shown were incorrect, we revised accordingly based on bias and RMSE listed in Table 6: "[…] shows a decreased bias from 606 mm w.e. in RACMO2.3p1 to 424 mm w.e. in version 2.3p2, and reduced RMSE from -133 mm w.e. to -54 mm w.e., and an increased R2 from 0.92 to 0.97."

P. 12, Lines 358-362: Is this correction applied to the values in Fig. 9?

This correction is applied to both RACMO2 versions in Figs. 9a and c. It is now mentioned in the figure caption as: "At S10, modelled SMB is estimated as the difference between total precipitation and melt.".

P. 13, Lines 402-405: Provide some numbers to illustrate that the new version performs as well as the previous version.

We included the following numbers: "[…] as well as the previous version, i.e. bias of 1.20 m w.e. yr-1 and RMSE of 0.47 m w.e. yr-1 (Noël et al., 2016), although […]".

P. 13, Lines 419-420: What are the new values for RMSE, bias, and error at QAS_L?

We calculated a model (RACMO2.3p2) RMSE and bias of 2.35 m w.e. and 2.21 m w.e. at QAS_L, respectively. This is now mentioned in the revised manuscript.

Tables 1-5: The term ME is used here for melt energy, but the term M is used in the text. These should be consistent. As noted in the general comments I believe this should really refer to the net energy balance. Captions for Tables 2-5 can be reduced to "Same as Table 1 for Station…"

Please, see our previous response in the General comments. We replaced ME in Tables 2-6 by M.

Figure 11: The red points in (a) indicate something different from (b) and (c). I feel that the authors should include RACMO2.3p2 for (a), (b) and (c), and use the same color scheme. A different color could be used to show the measurements from QAS_L.

We judge that showing RACMO2.3p1 data in Fig. 11b and c would make the scatter plots unclear and confusing. A similar comparison using RACMO2.3p1 data is already conducted in Noël et al. (2016). We now display QAS_L data in orange and we modified the caption accordingly.

Units for statistics should be the same for all figures if possible, and should correspond to the units in the text.

We deem that units should remain mm w.e. in Fig. 11a as bias and RMSE are relatively small. However, we think that m w.e. is more adequate for Figs. 11 b and c due to the larger bias and RMSE (2 orders of magnitude). Therefore, we decided not to change the units in the revised manuscript.

Also, I believe the third line of caption: "version 2.3 (red)" should read "version 2.3p1 (red)".

Done.

**Technical Corrections**

P. 1, Line 10: Change "than the previous model version" to "compared with the previous model version".
OK.

P. 2, Line 29: Change "to explicitly resolve" to "of explicitly resolving". OK.

P. 2, Line 33: Change "evaluate and improve" to "evaluating and improving". OK.

P. 2, Line 45:". If the authors are still referring to previous versions of the model, change to "is underestimated" to "was underestimated".

We kept "is underestimated" as this is still valid in version 2.3p2.

P. 2, Line 50: Change "near-kilometre" to "near-kilometre-scale" We replaced by "near-kilometre scale".

P. 3, Line 55: Change "all over Greenland." to "across the GrIS." OK.

P. 5, Line 132: Add "an" before "11 km horizontal…" OK.

P. 5, Lines 134-135: Mention that the model domain is shown in Fig. 1 to make clear what Fig. 1 is showing. We reformulated as: "[…] 6-hourly basis over the model domain shown in Fig. 1.".

P. 7, Line 191: Change "of 23 AWS" to "from 23 AWS" OK.

P. 7, Line 192: Change "output" to "outputs". OK.

P. 7, Line 194: Add "and" after "10-m wind speed," OK.

P. 7, Line 201: change "with a small negative bias" to "with the model exhibiting a small negative bias" for clarity. OK.

P. 8, Line 231 and Line 239: Change "too low cloud cover" to "underestimated cloud cover". OK.

P. 8, Line 241: The van den Broeke (2008) reference seems to be missing from the reference list. This reference is actually Smeets and van den Broeke (2008). This has been corrected.

P. 9, Line 258: Change "too large SHF" to "SHF to be overestimated". OK.

P. 10, Lines 295 – 298: The language could be improved here. Suggested revision: "In Section 3, we discussed the overall good ability of RACMO2.3p2 to reproduce the contemporary climate of the GrIS, which is essential for estimating realistic SMB patterns. Here we compare SMB from RACMO2.3p2 and RACMO2.3p1 over the GrIS. For further evaluation, we focus on three regions where there are large differences in SMB between the two versions." Thank you, we revised accordingly.

P. 11, Line 318: Change "very GrIS margins" to "extreme margins of the GrIS". OK.

P. 12, Lines 360-362: This sentence is a bit wordy… suggest changing "decreasing the bias…" to "decreasing the bias by 260 mm w. e. yr-1 to -40 mm w. e. yr-1 and the RMSE by 200 mm w. e. yr-1 to 210 mm w. e. yr-1." OK, thank you.

P. 12, Line 378: Change "3 months" to "3 month". OK.

P. 14, Line 426: Add "an" before "11 km resolution". OK.

P. 14, Line 434: change "narrow ablation zones" to "the narrow ablation zone". OK.

P. 14, Line 444: Change "to capture" to "to capturing". OK.

P. 15, Line 455: Change "cryoconites" to "cryoconite". OK.

P. 15, Line 467: Change "proves to accurately capture" to "accurately captures". OK.

Figure 2: Although it is not necessary since the caption provides a description, a legend on Fig. 2a would be useful for the reader. Done.

Figure 5: The dashed lines on the legend are hard to distinguish from solid lines. Although relatively small, we judge that Fig. 5 is readable as is. We did not modify this figure in the revised manuscript.

Figure 9: The black and blue colors are a bit hard to distinguish. Can the blue color be made slightly brighter? In caption, remove "the" in "for the S5". We decided to display observed data in light gray, and kept dark blue for RACMO2.3p2 data to remain consistent with other figures. Thank you.

Figure 10: The yellow line is difficult to see. The color could be made slightly darker. Add "and" after "a) daily snow albedo". As in Fig. 9, observational data are now displayed in light gray.

Fix references to follow format for The Cryosphere. We hope that the references are now suitable.

**Reviewer #2:**

This is a very nice well-written and detailed paper describing the revised RACMO2.3p2 model and the effect of improved model tuning on the Greenland ice sheet surface mass balance estimates. This kind of paper is extremely important for users of SMB data to read and digest in order to understand the likely biases and uncertainties within model output and the thorough analysis, while not really presenting much novel scientific research is an important addition to the scientific canon. It is very well structured and easy to read and the authors are to be congratulated on a thorough overview. That said I have some issues, which I feel should be addressed before final publication.

**Points for consideration**

In section 2.3 Model Updates, it is noted that there have been some changes to the cloud scheme but these are not discussed in much detail and it is not clear how the large adjustments to the lcrit in mixed phase and ice clouds were reached. This is a pretty serious adjustment of the model as many of the other model parametrizations in the radiation scheme are likely to be tuned to these kind of values, possibly giving erroneous results or different compensating biases. However little detail is given as to how or why the particular values for these adjustments were chosen. Nor are the effects of this adjustment alone described in any detail – for instance on lines 398 to 400 reference is made to a precipitation bias in the SE compared to observations, but it is not clear if this bias is reduced or increased from RACMO2.3p1 and if this is a result of the cloud scheme changes or for example the change in topography caused by moving to the GIMP DEM. More information on how this change in cloud parametrizations has altered precipitation in particular would be helpful, as the differences in the topography shown in figure 2 seem also to be related to the change in distribution of precipitation, at least in some locations as shown in figure 7.

We agree that the cloud scheme updates, i.e. tuning of the critical cloud content (lcrit), is not described in sufficient detail. The goal of this tuning was to obtain a uniform increase in precipitation over Greenland in order to minimize the inland dry bias observed in RACMO2.3p1.

To that end, we carried out a series of sensitivity experiments, i.e. one-year simulations, to test changes in spatial distribution as well as in the time scale of precipitation formation. From these experiments, we found a linear relationship between the critical cloud content (lcrit) for mixed and ice clouds, the vertical integrated cloud content (liquid and ice water paths that also affect the SEB through changes in cloud optical thickness), and the integrated precipitation over Greenland. Increasing lcrit for mixed (2x) and ice clouds (5x) uniformly enhances precipitation, except for some high accumulation regions in southeast Greenland (see Section 5.1 and Fig. 11a).

These new settings were then tested for a longer period. This approximately cancelled the dry bias observed in RACMO2.3p1. This also led to larger but realistic vertical integrated cloud content and did not strongly affect the SEB and surface climate of the GrIS. Note that lcrit is also strongly adjusted in the ECMWF physics compared to commonly used values in the literature. For example, for snow clouds modelled by a cloud parcel model, Lin et al. (1983) set lcrit to $1^{.}10^{-3}$ kg kg$^{-1}$, while the ECMWF physics uses a value of $0.3^{.}10^{-4}$ kg kg$^{-1}$. The justification for this is provided at p. 90 of ECMWF-IFS (2008): "A lower value (of lcrit) is appropriate for a GCM sized grid box (unless subgrid cloud variability is explicitly taken into account)". As RACMO2 employs much smaller grid boxes than GCMs, higher values of lcrit, typically $\sim 10^{-4}$ kg kg$^{-1}$, are not unreasonable.

In brief, changes in precipitation shown in Fig. 7a are mostly driven by the tuning of lcrit. Locally, large changes in mountainous areas, i.e. notably in southeast Greenland, are due to changes in the topography. Likewise, tuned lcrit is the main driver of changes in SWd and LWd through changes in cloud optical thickness, which of course slightly impact the other SEB components. However, these are even more affected by other changes applied in the new model version, i.e. changes in ice and snow albedos. To clarify this, we added the following sentences: "The values of lcrit adopted in RACMO2 were obtained

after conducting a series of sensitivity experiments, i.e. one-year simulations, to test the dependence of precipitation formation efficiency, spatial distribution and cloud moisture content on lcrit and other cloud tuning parameters. From these experiments, we found a linear relationship between lcrit for mixed and ice clouds, the vertical integrated cloud content, i.e. liquid and ice water paths that also affect the SEB through changes in cloud optical thickness, and the integrated precipitation over Greenland. These new settings were then tested for a longer period and proved to almost cancel the dry bias observed in RACMO2.3p1 (see Section 5.1). This led to larger but realistic vertical integrated cloud content and did not strongly affect the SEB and surface climate of the GrIS. For instance, the induced changes of surface downward shortwave and longwave radiation are only about -4 W m$^{-2}$ and 7 W m$^{-2}$, respectively, peaking in central Greenland. While the obtained increase in lcrit is relatively large, especially for ice clouds, it is important to note that it is also strongly adjusted in the original ECMWF physics compared to commonly used values in the literature: e.g. Lin et al. (1983) set lcrit to $1 \cdot 10^{-3}$ kg kg$^{-1}$ for ice clouds, while the ECMWF physics, tuned for GCM sized grid cells, uses a value of $0.3 \cdot 10^{-4}$ kg kg$^{-1}$ (ECMWF-IFS, 2008). As lcrit depends on model grid resolution, i.e. GCMs running at lower spatial resolution require lower value of lcrit (ECMWF-IFS, 2008), the use of a larger lcrit for e.g. ice clouds ($1.5 \ 10^{-4}$ kg kg$^{-1}$) in RACMO2 is deemed reasonable. In addition, this value remains well within the range of values previously presented in the literature (Lin et al., 1983)."

On a similar theme, I note that the small improvement in LWd and SWd on the K-transect is reasonably attributed to the change in the cloud scheme. It would be very interesting to see if this improvement is consistent across Greenland at stations other than the K-transect. There is some reason to believe that western Greenland is often modelled well but in other regions RCMs do a less good job of reproducing observed climate variables, possibly due to biases in cloud schemes as well as the complex topography in other areas. As there is now a fairly large amount of data available from Promice stations it would be nice to see some geographical spread in the figures presented in figure 5 and tables 1-5, perhaps limited to maybe 3-4 extra stations in north, south and east Greenland to determine if the positive results from western Greenland are replicated elsewhere.

To provide a regional evaluation of meteorological and radiation data, we decided to add individual scatterplots for four different sectors of the ice sheet (NE [Figs. S3-4], NW [Figs. S5-6], SE [Figs. S7-8], SW [Figs. S9-10]). These scatterplots are similar to Figs. 3 and 4. They list statistics for the four regions, respectively, and clearly stress that RACMO2.3p2 agrees as well with observations in these four sectors. See also our response to the General comments 2) of reviewer #1. We also added the following sentences in the revised manuscript; in Section 3.1: "To provide some regional insight on the model performance, Table S1 and Figs. S1-S4 compare modelled meteorological data from RACMO2.3p2 with AWS measurements (green dots in Fig. 1) clustered in four sectors of the GrIS, i.e. NW, NE, SW and SE, respectively. These sectors correspond to the four quadrants delimited by longitude 40ºW and latitude 70ºN, respectively. These regional scatter plots unambiguously show that RACMO2.3p2 performs as good in each of these four sectors of the GrIS.".

In Section 3.2: "Clustering AWS measurements within four sectors of the GrIS (Figs. S6-S9 and Table S1), RACMO2.3p2 shows good and equivalent agreement in NW, NE, SW and SE Greenland, respectively.".

Upper atmosphere relaxation is mentioned on line 136-7 but no details are given. I would like to see this expanded with details on which fields are nudged and at which levels in the atmosphere as this is important for interpreting the atmospheric model output.
We elaborated as follows: "To better capture SMB interannual variability in this new model version, upper atmosphere relaxation (UAR or nudging) of temperature and wind fields is applied every 6 hours for model atmospheric levels above 600 hPa, i.e. ~ 4 km a.s.l. (Van de Berg and Medley [2016]). UAR is not applied to atmospheric humidity fields in order not to alter clouds and precipitation formation in RACMO2.".

The authors acknowledge that boundary forcing is important for results (line 20) but the differences between results from ERA40 and ERA-Interim forced years are not explored at all. It would be

helpful to have a time series of SMB and the components for the full 1958-2015 averaged over the full ice sheet for the full period. This would show if, for example, there is a marked change in precipitation or melt potentially resulting from the switch in boundary forcing in 1979 is visible in more detail.

Time series of RACMO2.3p1 have been already published in e.g. Van den Broeke et al. (2016) [11 km] and Noël et al. (2017) [1 km] and show no abrupt changes (larger than the interannual variability) between 1978 and 1979, i.e. when the re-analysis forcing switches from ERA-40 to ERA-Interim. The same also applies to RACMO2.3p2 simulation.

It would also give a better sense of the interannual and decadal scale variability in SMB of Greenland. Plotting these with model version p1 masked with the same ice mask would also allow us to assess the differences in SMB over the full ice sheet that result from the improvements introduced here.

As discussed in Section 5.1, trends and time series cannot be directly derived from the new RACMO2.3p2 version as "correct" precipitation and underestimated runoff, i.e. due to unresolved high melt rates over low-lying marginal outlet glaciers and narrow ablation zone, lead to overestimated GrIS SMB at 11 km resolution, highlighting the need for further statistical downscaling. In addition, relevant climatological averages (1958-2016) for the main SMB components are already listed in Sections 5.1 [11 km] and 5.2 [1 km]. At the discretion of the editor, we are happy to include time series of annual downscaled SMB components from RACMO2.3p2 at 1 km.

I am not quite clear if the improvements to the snow module are part of the online RACMO model or the offline firn model – I assume the former, but please clarify this in sections 2.1 and 2.3.

RACMO2 (climate model including a snow/firn module) and FDM-IMAU (firn model) are two different models. As mentioned, IMAU-FDM is run offline and is forced by the climate data of RACMO2. In Section 2.1 and 2.3, we only refer to RACMO2 settings and updates and do not discuss FDM-IMAU. FDM-IMAU is only mentioned in Section 2.1 to clarify how a snowpack initialization has been obtained for the 1[st] of September 1957, date at which the RACMO2.3p2 simulation starts. FDM-IMAU simulations forced by RACMO2.3p2 will be discussed in a forthcoming paper. See also our response to reviewer #1.

"The model has about 40 active […] 1957 using estimates of temperature and density profiles derived by the offline […] (Ligtenberg et al., 2011). These profiles are obtained by repeatedly running IMAU-FDM over 1960-1979 forced by the outputs of the previous RACMO2.3p1 climate simulation until the firn column reaches an equilibrium."

On line 216 you note that AWS data is sometimes spurious, Ryan et al 2017 in GRL also showed that the siting of stations (for very good reasons!) also leads to spurious underestimation of albedo – this should probably be referenced. This reference has been included in the revised manuscript. Thank you.

Manuscript prepared for J. Name
with version 5.0 of the LATEX class copernicus.cls.
Date: 11 January 2018

**Modelling the climate and surface mass balance of polar ice sheets using RACMO2, Part 1: Greenland (1958-2016)**

Brice Noël[1], Willem Jan van de Berg[1], J. Melchior van Wessem[1], Erik van Meijgaard[2], Dirk van As[3], Jan T. M. Lenaerts[4], Stef Lhermitte[5], Peter Kuipers Munneke[1], C. J. P. Paul Smeets[1], Lambertus H. van Ulft[2], Roderik S. W. van de Wal[1], and Michiel R. van den Broeke[1]

[1]Institute for Marine and Atmospheric research Utrecht, University of Utrecht, Utrecht, Netherlands.
[2]Royal Netherlands Meteorological Institute, De Bilt, Netherlands.
[3]Geological Survey of Denmark and Greenland (GEUS), Copenhagen, Denmark.
[4]Department of Atmospheric and Oceanic Sciences, University of Colorado, Boulder, USA.
[5]Department of Geoscience & Remote Sensing, Delft University of Technology, Delft, Netherlands.

*Correspondence to:* Brice Noël (B.P.Y.Noel@uu.nl)

**Abstract.**

We evaluate modelled Greenland ice sheet (GrIS) near-surface climate, surface energy balance (SEB) and surface mass balance (SMB) from the updated regional climate model RACMO2 (1958-2016). The new model version, referred to as RACMO2.3p2, incorporates updated glacier outlines, topography and ice albedo fields. Parameters in the cloud scheme governing the conversion of cloud condensate into precipitation have been tuned to correct inland snowfall underestimation; snow properties are modified to reduce drifting snow and melt production in the ice sheet percolation zone. The ice albedo prescribed in the updated model is lower at the ice sheet margins, increasing ice melt locally. RACMO2.3p2 shows good agreement compared to in situ meteorological data and point SEB/SMB measurements, and better resolves the spatial patterns and temporal variability of SMB compared with the previous model version, notably in the northeast, southeast, and along the K-transect in southwestern Greenland. This new model version provides updated, high-resolution gridded fields of the GrIS present-day climate and SMB, and will be used for projections of the GrIS climate and SMB in response to future climate scenario in a forthcoming study.

**Keywords.** RACMO, SMB, SEB, Greenland

**1 Introduction**

Predicting future mass changes of the Greenland ice sheet (GrIS) using regional climate models (RCMs) remains challenging (Rae et al., 2012). The reliability of projections depend on the ability of RCMs to reproduce the contemporary GrIS climate and surface mass balance (SMB), i.e. snowfall accumulation minus ablation from meltwater runoff, sublimation and drifting snow erosion (Van Angelen et al., 2013a; Fettweis et al., 2013). In addition, RCM simulations are affected by the quality of the re-analysis used as lateral forcing (Fettweis et al., 2013, 2017; Bromwich et al., 2015) and by the accuracy of the ice sheet mask and topography prescribed in models (Vernon et al., 2013).

Besides direct RCM simulations, the contemporary SMB of the GrIS has been reconstructed using various other methods, e.g. Positive Degree Day (PDD) models forced by statistically downscaled re-analyses (Hanna et al., 2011; Wilton et al., 2016), mass balance models forced by the climatological output of an RCM (HIRHAM4) (Mernild et al., 2010, 2011), and reconstruction of SMB obtained by combining RCM outputs with temperature and ice core accumulation measurements (Box, 2013). In addition, Vizcaíno et al. (2013) and Cullather et al. (2014) respectively used the Community Earth System Model (CESM) at 1° resolution (∼100 km) and the Goddard Earth Observing System model version 5 (GEOS-5) at 0.5° resolution (∼50 km) to estimate recent and future mass losses of the GrIS.

[revised manuscript text omitted]

160 **2.4 Initialisation and set up**

To enable a direct comparison with previous runs, RACMO2.3p2 is run at an 11 km horizontal resolution for the period 1958-2016, and is forced at its lateral boundaries by ERA-40 (1958-1978) (Uppala et al., 2005) and ERA-Interim (1979-2016) (Dee et al., 2011) re-analyses on a 6-hourly basis over the model domain shown in Fig. 1. The forcing consists of temperature, specific humidity,
165 pressure, wind speed and direction being prescribed at each of the 40 vertical atmosphere hybrid model levels. To better capture SMB inter-annual variability in this new model version, upper atmosphere relaxation (UAR or nudging) of temperature and wind fields is applied every 6 hours for model atmospheric levels above 600 hPa, i.e. $\sim$4 km a.s.l. (Van de Berg and Medley, 2016). UAR is not applied to atmospheric humidity fields in order not to alter clouds and precipitation formation
170 in RACMO2. As the model does not incorporate a dedicated ocean module, sea surface temperature and sea ice cover are prescribed from the re-analyses (Fiorino, 2004; Stark et al., 2007). The model has about 40 active snow layers that are initialised in September 1957 using estimates of temperature and density profiles derived from the offline IMAU Firn Densification Model (IMAU-FDM) (Ligtenberg et al., 2011). These profiles are obtained by repeatedly running IMAU-FDM over 1960-1979
175 forced by the outputs of the previous RACMO2.3p1 climate simulation until the firn column reaches an equilibrium. The data spanning the winter season up to December 1957 serve as an additional spin up for the snowpack and are therefore discarded in the present study.

Relative to previous versions, the integration domain extends further to the west, north and east (Fig. 1). This brings the northernmost sectors of the Canadian Arctic Archipelago and Svalbard well
180 inside the domain interior, and further away from the lateral boundary relaxation zone (24 grid cells, black dots in Fig. 1). In addition, RACMO2.3p2 utilises the 90-m Greenland Ice Mapping Project (GIMP) Digital Elevation Model (DEM) (Howat et al., 2014) to better represent the glacier outlines and the surface topography of the GrIS. Compared to the previous model version, which used the 5 km DEM presented in Bamber et al. (2001), the GrIS area is reduced by 10,000 km$^2$ (Fig. 2a). This
185 mainly results from an improved partitioning between the ice sheet and peripheral ice caps, for which

the ice-covered area has, in equal amounts, decreased and increased, respectively. In RACMO2, a grid-cell with an ice fraction $\geq 0.5$ is considered fully ice-covered. The updated topography shows significant differences compared to the previous version, especially over marginal outlet glaciers where surface elevation has considerably decreased (Fig. 2b). Bare ice albedo is prescribed from the

190   500 m MODerate-resolution Imaging Spectroradiometer (MODIS) 16-day Albedo version 5 product (MCD43A3v5), as the 5% lowest surface albedo records for the period 2000-2015 (vs. 2001-2010 in older versions; Fig. 2c). In RACMO2, minimum ice albedo is set to 0.30 for dark ice in the low-lying ablation zone, and a maximum value of 0.55 for bright ice under perennial snow cover in the accumulation zone. In previous RACMO2 versions, bare ice albedo of glaciated grid cells without

195   valid MODIS estimate were set to 0.47 (Noël et al., 2015).

**2.5 Observational data**

To evaluate the modelled contemporary climate and SMB of the GrIS, we use daily average meteorological records of near-surface temperature, wind speed, relative humidity, air pressure and down/upward short/longwave radiative fluxes, retrieved from 23 AWS for the period 2004-2016

200   (green dots in Fig. 1). Erroneous radiation measurements, caused e.g. by sensor riming, were discarded by removing daily records showing $\mathrm{SW}_{d\ bias} > 6\ \sigma_{bias}$, where $\mathrm{SW}_{d\ bias}$ is the difference between daily modelled and observed $\mathrm{SW}_d$ and $\sigma_{bias}$ is the standard deviation of the daily $\mathrm{SW}_d$ bias for all measurements. In addition, measurements affected by sensor heating in summer, i.e. showing $\mathrm{LW}_u > 318\ \mathrm{W\ m}^{-2}$, were eliminated as these values represent $\mathrm{T}_s > 0°\mathrm{C}$ for $\epsilon \approx 0.99$, where $\mathrm{T}_s$ is

205   the surface temperature and $\epsilon$ the selected emissivity of snow or ice. We only used daily records that were simultaneously available for each of the four radiative components. Eighteen of these AWS sites are operated as part of the Programme for Monitoring of the Greenland Ice Sheet (PROMICE, www.promice.dk) covering the period 2007-2016 (Van As et al., 2011). Four other AWS sites, namely S5, S6, S9 and S10 (2004-2016), are located along the K-transect in southwest Greenland

210   (67°N, 47-50°W) (Smeets et al., 2017). Another AWS (2014-2016) is situated in southeast Greenland (66°N; 33°W) at a firn aquifer site (Forster et al., 2014; Koenig et al., 2014). The latter five sites are operated by the Institute for Marine and Atmospheric research at Utrecht University (IMAU).

We also use in situ SMB measurements collected at 213 stake sites in the GrIS ablation zone (yellow dots in Fig. 1; Machguth et al. (2016)) and at 182 sites in the accumulation zone (white dots

215   in Fig. 1) including snow pits, firn cores (Bales et al., 2001, 2009), and airborne radar measurements (Overly et al., 2016). We exclusively selected measurements that temporally overlap with the model simulation (1958-2016). To match the observational period, daily modelled SMB is cumulated for the exact number of measuring days at each site.

For model evaluation, we select the grid cell nearest to the observation site in the accumulation

220   zone. In the ablation zone, an additional altitude correction is applied by selecting the model grid cell with the smallest elevation bias among the nearest grid cell and its eight adjacent neighbours.

One ablation site and seven PROMICE AWS sites presented an elevation bias in excess of $> 100$ m compared to the model topography and were discarded from the comparison.

In addition, we compare modelled SMB with annual glacial ice discharge (D) retrieved from the combined Zachariae Isstrøm and Nioghalvfjerdsbrae glacier catchments in northeast Greenland (1975-2015; yellow line in Fig. 6a), presented in Mouginot et al. (2015).

**3 Results: near-surface climate and SEB**

We evaluate the modelled present-day near-surface climate of the GrIS in RACMO2.3p2 using data from 23 AWS sites (see Section 2.5). Then, we discuss in more detail the model performance at 4 AWS along the K-transect and compare RACMO2.3p2 outputs to those of RACMO2.3p1.

**3.1 Near-surface meteorology**

Figure 3 compares daily mean values of 2-m temperature, 2-m specific humidity, 10-m wind speed, and air pressure collected at 23 AWS sites with RACMO2.3p2 output. The modelled 2-m temperature is in good agreement with observations ($R^2 = 0.95$) and with a RMSE of $\sim 2.4°C$ and a small cold bias of $\sim 0.1°C$ (Fig. 3a). As specific humidity is not directly measured at AWS sites, it is calculated from measured temperature, pressure and relative humidity following Curry and Webster (1999). The obtained 2-m specific humidity is accurately reproduced in the model ($R^2 = 0.95$) with a RMSE $\sim 0.35$ g kg$^{-1}$ and a negative bias of 0.13 g kg$^{-1}$ (Fig. 3b). The same holds for daily records of 10-m wind speed ($R^2 = 0.68$; Fig. 3c), with the model exhibiting a small negative bias and RMSE of $\sim 2$ m s$^{-1}$. Surface pressure is also well represented ($R^2 = 0.99$) with a small negative bias of 0.8 hPa and RMSE $< 6$ hPa (Fig. 3d). A systematic pressure bias at some stations results from the (uncorrected) elevation difference with respect to the model, which can be as large as 100 m. To provide some regional insight on the model performance, Table S1 and Figs. S1-S4 compare modelled meteorological data from RACMO2.3p2 with AWS measurements (green dots in Fig. 1) clustered in four sectors of the GrIS, i.e. NW, NE, SW and SE, respectively. These sectors correspond to the four quadrants delimited by longitude 40°W and latitude 70°N, respectively. These regional scatter plots unambiguously show that RACMO2.3p2 performs as good in each of these four sectors of the GrIS.

Table 1 and Fig. S5 compare the agreement of RACMO2.3p2 and version 2.3p1 with in situ measurements. We find an overall improvement in the updated model version, showing a smaller bias and RMSE as well as an increased variance explained. Notably, the remaining negative bias in 2-m temperature (Fig. S5a) and the systematic dry bias (Fig. S5b) in RACMO2.3p1 have almost vanished in the updated model version (Figs. 3a and b).

**3.2 Radiative fluxes**

Figure 4 shows scatter plots of modelled and measured daily mean radiative fluxes, i.e. short/longwave down/upward radiation. Radiative fluxes are also well reproduced by the model with $R^2$ ranging from 0.83 for $LW_d$ to 0.95 for $SW_d$ (Fig. 4), showing relatively small biases of -7.1 W m$^{-2}$ and 3.8 W m$^{-2}$, and RMSE of 21.2 W m$^{-2}$ and 27.1 W m$^{-2}$, respectively. The negative biases in $LW_d$ and 2-m temperature partly lead to $LW_u$ underestimation of 4.4 W m$^{-2}$ with a small RMSE of 12.1 W m$^{-2}$, in combination with positive bias in $SW_d$ suggests an underestimation of cloud cover in the ice sheet marginal regions, where most stations are located. The larger bias and RMSE in $SW_u$ of 6.8 W m$^{-2}$ and 32.1 W m$^{-2}$, respectively, can be ascribed to overestimated surface albedo, especially during summer snowfall episodes, when a bright fresh snow cover is deposited over bare ice. In RACMO2, precipitation falls vertically, i.e. no horizontal transport is allowed, and is assumed to be instantly deposited at the surface. Consequently, the spatial distribution of summer snow patches may be locally inaccurate, resulting in large albedo discrepancies when compared to point albedo measurements. Note that these AWS radiation measurements are also prone to potentially large uncertainties due to preferred location on ice hills, sensor tilt, riming and snow/rain deposition on the instruments, leading to spurious albedo and $SW_u$ data (Ryan et al., 2017), e.g. the upper left dots in Fig. 4b. Clustering AWS measurements within four sectors of the GrIS (Figs. S6-S9 and Table S1), RACMO2.3p2 shows good and equivalent agreement in NW, NE, SW and SE Greenland, respectively.

Compared to the previous model version (Table 1), changes in the cloud scheme have significantly improved the representation of $LW_d$ (Figs. 4c and S10c), showing a reduced negative bias and RMSE. These modifications have also somewhat decreased the positive bias in $SW_d$ (Fig. 4a), relative to RACMO2.3p1 (Fig. S10a). In addition, $LW_u$ is notably improved in RACMO2.3p2: the remaining negative bias in $LW_u$ has almost vanished (Figs. 4d and S10d). This can be partly explained by the much better resolved 2-m temperature in RACMO2.3p2.

**3.3 Seasonal SEB cycle along the K-transect**

[revised manuscript text omitted]

**4.2 Northeast Greenland**

For northeast Greenland's two main glaciers, Zachariae Isstrøm and Nioghalvfjerdsbrae (79N glacier;
385 yellow line in Fig. 6a), solid ice discharge (D) estimates are available for the period 1975-2015 (Mouginot et al., 2015). Assuming that this glacier catchment draining $\sim$12% of the GrIS area remained in approximate balance until $\sim$2000 (Mouginot et al., 2015), i.e. D $\approx$ SMB, measurements

of D at the grounding line of these marine-terminating glaciers can be used to evaluate modelled SMB.

In these two catchments, model updates significantly improve the representation of SMB, that was substantially underestimated in the previous version. Figure 8a compares ice discharge (black dots) with modelled SMB (RACMO2.3p2 as blue dots and 2.3p1 in red) integrated over the two glacier basins for 1958-2015. In a balanced system, i.e. before discharge accelerated in 2001, SMB equals ice discharge. Averaged over 1975-2001, modelled SMB in RACMO2.3p2 (20.5 Gt yr$^{-1}$) is similar to the estimated glacial discharge of 21.2 Gt yr$^{-1}$, significantly improving upon version 2.3p1 (15.8 Gt yr$^{-1}$). The negative bias in RACMO2.3p2 (0.7 Gt yr$^{-1}$; dashed blue line) is reduced by almost a factor of eight relative to the previous version (5.4 Gt yr$^{-1}$) and SMB now equals discharge within the uncertainty. However, it is important to note that, while good agreement is obtained between averaged SMB and D before 2001, suggesting a glacier catchment in approximate balance as in Mouginot et al. (2015), this does not necessarily confirm that spatial and temporal variability of northeast Greenland SMB is accurately resolved by the model. Averaged over 2001-2015, basin mass loss accelerated due to enhanced surface runoff, decreasing SMB by 4.2 Gt yr$^{-1}$, and increased ice discharge (2.8 Gt yr$^{-1}$).

Figures 8b and c show mean SMB for 1958-2015 as modelled by RACMO2.3p2 and 2.3p1, respectively. In the percolation zone, the difference between the two model versions primarily results from the smaller refrozen snow grain size that reduces melt and runoff through increased surface albedo in RACMO2.3p2. To a smaller extent, reduced soot concentration delays the onset of melt in summer. In the ablation zone, snow cover persists longer before bare ice is exposed in late summer, in turn reducing runoff (Fig. 7d). Superimposed on this, precipitation has increased over the whole glacier basin (Fig. 7a), allowing for enhanced refreezing in snow (Fig. 7f) hence increasing SMB by 4.7 Gt yr$^{-1}$ in RACMO2.3p2 (Fig. 6b). Note the large inter-annual variability in modelled SMB showing a maximum and minimum value of approximately 30 Gt yr$^{-1}$ and 8.5 Gt yr$^{-1}$ in RACMO2.3p2 vs. 25 Gt yr$^{-1}$ and 0 Gt yr$^{-1}$ in the previous version, stressing the importance of accurately modelling individual SMB components. In this dry region, underestimation of snowfall accumulation in RACMO2.3p1 initiated a pronounced feedback decreasing SMB: active drifting snow processes erode the shallow snow cover, exposing bare ice prematurely and moving the equilibrium line too far inland (Figs. 8b and c).

**4.3 K-transect**

The K-transect in southwest Greenland consists of eight stake sites where SMB is measured annually (yellow dots in Fig. 6a) (Van de Wal et al., 2012; Machguth et al., 2016). Figure 9a compares modelled (RACMO2.3p2 as blue dots and RACMO2.3p1 in red), with observed SMB (black dots) along the transect, averaged for the period 1991-2015. Using mean annual SMB at each station, the updated model shows a decreased bias from 606 mm w.e. in RACMO2.3p1 to 424 mm w.e. in

version 2.3p2, and reduced RMSE from -133 mm w.e. to -54 mm w.e., and an increased $R^2$ from

[revised manuscript text omitted]

Manuscript prepared for J. Name
with version 5.0 of the LaTeX class copernicus.cls.
Date: 11 January 2018

[revised manuscript text omitted]

**Table S1.** Difference between daily modelled RACMO2.3p2 (2004-2016) and observed meteorological data and SEB components collected at 23 AWS (green dots in Fig. 1) and clustered within four GrIS sectors: NW (>40°W, >70°N; 4 AWS), NE (<40°W, >70°N; 2 AWS), SE (<40°W, <70°N; 4 AWS) and SW (>40°W, <70°N; 13 AWS) Greenland. Statistics include model bias (RACMO2.3p2 - observations), RMSE of the bias, the determination coefficient of daily mean data as well as the percentage of measurements located in each GrIS sector. All fluxes are set positive.